

# Impact of updated radiative transfer scheme in RACMO2.3p3 on the surface mass and energy budget of the Greenland ice sheet

Christiaan T. van Dalum[1], Willem Jan van de Berg[1], and Michiel R. van den Broeke[1]

[1]Institute for Marine and Atmospheric Research, Utrecht University, Utrecht, The Netherlands

**Correspondence:** Christiaan van Dalum (c.t.vandalum@uu.nl)

**Abstract.** This study evaluates the impact of a new snow and ice albedo and radiative transfer scheme on the surface mass and energy budget for the Greenland ice sheet in the latest version of the regional climate model RACMO2, version 2.3p3. We also evaluate the modeled (sub)surface temperature and snow melt, as subsurface heating by radiation penetration now occurs. The results are compared to the previous model version and are evaluated against stake measurements and automatic weather station data of the K-transect and PROMICE projects. In addition, subsurface snow temperature profiles are compared at the K-transect, Summit and southeast Greenland. The surface mass balance is in good agreement with observations, and only changes considerably with respect to the previous RACMO2 version around the ice margins and in the percolation zone. Snow melt and refreezing, on the other hand, are changed more substantially in various regions due to the changed albedo representation, subsurface energy absorption and melt water percolation. Internal heating leads to considerably higher snow temperatures in summer, in agreement with observations, and introduces a shallow layer of subsurface melt.

## 1 Introduction

The Greenland ice sheet (GrIS) has been losing mass at an accelerating pace in the last decade (Box and Colgan, 2013; Kjeldsen et al., 2015; Bevis et al., 2019; Shepherd et al., 2020). Both surface runoff and solid ice discharge have increased, enhancing mass loss (Bigg et al., 2014). The relative contribution of surface processes with respect to ice discharge, however, has recently increased considerably (Enderlin et al., 2014; Kjeldsen et al., 2015; Mouginot et al., 2019), augmenting the need to accurately model the surface mass balance (SMB) in regional and global climate models.

The SMB, which is the difference between precipitation and ablation, i.e., runoff, sublimation and drifting snow erosion, is highly variable in space and time. Snow and ice melt typically dominate the SMB around the margins of the GrIS, leading to extensive runoff and mass loss of up to 3 m water equivalent (w.e.) yr$^{-1}$, while snow fall dominates in the interior (Van den Broeke et al., 2016). In the interior, melt events do not necessarily lead to runoff, as a significant fraction of all melt water refreezes locally (Steger et al., 2017). The length and spatial extent of melt events vary from year to year, but record-breaking melt events like during the summer of 2012 and 2019 can bring melt to almost the entire ice sheet (Nghiem et al., 2012; Bennartz et al., 2013; Tedesco et al., 2013; Tedesco and Fettweis, 2020), inducing a darkening of the snow pack and a change in snow structure. As similar melt events are expected to become more common in a warmer climate (IPCC, 2013; Shepherd et al., 2020), it is therefore important to adequately resolve the individual SMB components. In-situ observations, however, lack





the required spatial coverage and temporal sampling to fully capture these events, while satellites cannot adequately quantify melt rates, and thus the use of climate models is required (Rae et al., 2012; Goelzer et al., 2013; Leeson et al., 2018; Alexander et al., 2019).

Regional climate models are able to explicitly resolve the GrIS SMB components and other atmospheric and surface processes, and are characterized by a high temporal (subhourly) and spatial resolution (up to 5 km) (Fettweis et al., 2017; Langen et al., 2017; Powers et al., 2017; Niwano et al., 2018; Noël et al., 2018b). The SMB components can be statistically downscaled to an even higher resolution of up to 1 km (Noël et al., 2019). RCMs are known to perform well for most of the ice sheet, but struggle to resolve topographically inhomogeneous regions (Leeson et al., 2018). These regions are often located close to the ice margin in west Greenland where significant mass loss occurs, illustrating the need to further improve SMB related parameterizations.

Surface melt rate is determined by the surface energy budget (SEB), i.e., the sum of radiative, turbulent and ground heat fluxes. Parameterizations of radiative fluxes can still be improved in RCMs, leading to better surface melt estimates (Fettweis et al., 2017). A considerable part of incoming shortwave radiation, however, penetrates through the surface, heating snow and ice layers below (Kuipers Munneke et al., 2009; Warren, 2019; He and Flanner, 2020). This is especially important if ice rather than snow is at or close to the surface. Radiation scattering is limited in these large-grained ice layers and shortwave radiation is therefore also absorbed below the surface, acting as a heat source. As internal heat is not as effectively dissipated as at the surface, subsurface energy absorption can lead to subsurface melting if the surface is close to or at the melting point.

The polar (p) version of the Regional Atmospheric Climate Model (RACMO2), has been used extensively to model large scale, ice-sheet wide developments of the Greenland (Noël et al., 2018b; Noël et al., 2019) and Antarctic ice sheets (Van de Berg and Medley, 2016; Van Wessem et al., 2018), but also for glaciated areas on a smaller scale, like the glaciers and ice caps of the Canadian Arctic (Noël et al., 2018a). Furthermore, RACMO2 has been used to investigate physical processes like the snowmelt-albedo feedback (Jakobs et al., 2019) and föhn winds (Wiesenekker et al., 2018). Radiation penetration is not included, but recent developments in snow albedo parameterizations, i.e., the fraction of incoming shortwave radiation that is reflected, and radiation transfer schemes now allow the implementation of a radiation penetration module in RACMO2 (Van Dalum et al., 2019). The new version RACMO2.3p3, henceforth Rp3, incorporates a new snow and ice albedo and radiative transfer scheme, which includes internal heating by radiation penetration, and updates to the firn module (Van Dalum et al., 2020).

This manuscript discusses the impact of the updated albedo and radiative transfer scheme and other model adjustments on the modeled SMB and SEB. Section 2 discusses the model and initialization, expands on the concepts of SMB, SEB and internal energy absorption, and discusses the in-situ data sets in more detail. In Section 3, results are analyzed by evaluating the SMB to observations and by comparing modeled SMB of Rp3 with the previous RACMO2 version, 2.3p2 (Noël et al., 2018b) (Rp2). Similar to Section 3, Section 4 focuses on the SEB. In Section 5, internal energy absorption due to radiation penetration is discussed. In Section 6, sensitivity experiments further highlight the impact of internal heating on the subsurface temperature and SMB. Finally, Section 7 summarizes the results and conclusions are drawn.





## 2 Methods and data

### 2.1 Regional climate model

The Regional Atmospheric Climate Model (RACMO2), developed at the Royal Netherlands Meteorological Institute (KNMI), couples the surface and atmospheric processes of the European Center for Medium-Range Weather Forecasts (ECMWF) Integrated Forecast System (IFS), cycle 33r1 (ECMWF, 2009), with the atmospheric dynamics of the High Resolution Limited Area Model, version 5.0.3 (HIRLAM, Undén et al., 2002). The polar (p) version of RACMO2, which is developed at the Institute for Marine and Atmospheric Research Utrecht (IMAU), is adjusted for glaciated areas by introducing a dedicated glaciated tile that includes snow and ice processes and more complete ice-atmosphere interaction (Noël et al., 2015).

Two major components have been updated in the new version Rp3 compared to the previous version Rp2: the multilayer firn module and the snow and ice albedo parameterization. More specifically, the multilayer firn module has increased the vertical resolution and the merging routine reduces mixing of layers with distinct characteristics, reducing numerical diffusion. Rp3 now typically has 50 to 60 active snow layers, with 100 as a maximum. Model output is only available for the first 20 layers and every 3 hours. The plane-parallel broadband snow albedo scheme based on Gardner and Sharp (2010) is replaced by the Two-streAm Radiative TransfEr in Snow Model (TARTES, Libois et al., 2013), which is coupled to RACMO2 with the Spectral-to-NarrOWBand ALbedo (SNOWBAL) module version 1.2 (Van Dalum et al., 2019). TARTES uses the asymptotic radiative transfer theory (Kokhanovsky, 2004; He and Flanner, 2020) and the radiative transfer equation (Jiménez-Aquino and Varela, 2005) to calculate a spectral albedo and subsurface energy absorption by using the geometric-optics method. SNOWBAL has been developed to couple the output of TARTES with the 14 contiguous shortwave spectral bands of the IFS physics scheme embedded in RACMO2, taking into account sub-band variations of both the albedo and the irradiance. Bands 13 and 14 are, however, excluded from calculations, as the albedo for these bands can be safely assumed to be zero (Gardner and Sharp, 2010) and all energy contributes to the SEB. Additionally, a new bare ice albedo parameterization using both TARTES and SNOWBAL has been developed. A more detailed model description and evaluation of the snow and ice albedo product can be found in Van Dalum et al. (2020).

### 2.2 Surface mass balance and surface energy budget

The surface energy budget (SEB) of RACMO2, with fluxes toward the surface defined as positive, is defined as

$$M = \mathrm{LW_d} + \mathrm{LW_u} + \mathrm{SW_d} + \mathrm{SW_u} + \mathrm{SHF} + \mathrm{LHF} + G_\mathrm{s}, \tag{1}$$

with $M$ the surface melt flux ($M = 0$ when surface temperature $T_\mathrm{s} < 273.15$ K), $\mathrm{LW_d}$, $\mathrm{LW_u}$, $\mathrm{SW_d}$ and $\mathrm{SW_u}$ the downward and upward longwave radiative fluxes and downward and upward shortwave radiative fluxes, respectively, SHF and LHF the sensible and latent heat fluxes and $G_\mathrm{s}$ the subsurface conductive heat flux, all in W m$^{-2}$. Melt water is allowed to percolate into





deeper layers using the tipping-bucket method, i.e., water fills a layer until the irreducible water saturation is reached (Coléou and Lesaffre, 1998). Excess water moves to the next unsaturated layer, where it can either refreeze, be retained or runs off.

Here, we adopt the following definition of the surface mass balance (SMB), in mm w.e. $\mathrm{yr}^{-1}$:

$$\mathrm{SMB} = \mathrm{SN} + \mathrm{RA} - \mathrm{SU} - \mathrm{ER} - \mathrm{RU}, \qquad (2)$$

with SN snowfall, RA rain, SU sublimation, ER drifting snow erosion and RU runoff. RU represents rain and meltwater that is not refrozen or retained in the firn layer. Strictly speaking, this definition of SMB represents the climatic mass balance, as internal accumulation is included (Cogley et al., 2011).

### 2.3 Internal energy absorption

In Rp3, the albedo and energy absorption profiles within snow layers are computed on full-radiation (FR) time steps, which is

every whole hour. For all other time steps, the albedo and absorption profiles of the previous FR time step are used as long as the sun is above the horizon. The internal energy absorption that is calculated by TARTES, however, is only valid on a FR time step, as snow layers and radiation entering the snow pack change within the hour.

To be able to calculate internal energy absorption at non-FR time steps, we assume that the net downward shortwave energy flux $F(z)$ decays exponentially within a model layer with attenuation length $\tau$, where depth $z > z_{\mathrm{top}}$, with $z_{\mathrm{top}}$ the depth of

the upper interface and $F_{\mathrm{top}}$ the net downward shortwave flux at the top of the snow layer. $F(z)$ is then:

$$F(z) = F_{\mathrm{top}} \mathrm{e}^{-(z - z_{\mathrm{top}})/\tau}. \qquad (3)$$

At each FR time step, $\tau$ is determined for each snow layer using Eq. (3) and the modeled absorption profile of TARTES. During non-FR time steps, these $\tau$'s are used to distribute the net absorbed shortwave radiation over the model layers. This procedure

is repeated for all spectral bands of Rp3, as both the albedo and $\tau$ depend on wavelength. For visible light, only a small fraction of incoming radiation is absorbed in the snow pack, but penetrates relatively deeply. For near-infrared (near-IR, 750–1400 nm) radiation, a large fraction is absorbed, but penetration and absorption is limited to the upper layers (Ebert et al., 1995; Gardner and Sharp, 2010; He et al., 2017, 2018; He and Flanner, 2020).

Due to the absorption behaviour for near-IR radiation, a major fraction of incoming radiation is absorbed in the upper

millimeters of the snow pack. As for this length scale, the time scale for heat diffusion to the surface is shorter than the model time step (4 minutes) and near surface shortwave radiation absorption therefore needs to be accounted for in the SEB. In order to distinguish between surface and internal energy absorption, we assume that the fraction of absorbed energy attributed to the SEB is 1 at the surface and linearly decreases to 0 at the maximum skin layer equilibration depth (SLED). Using scale analysis, we estimate the SLED for both snow and ice to be 5 mm (Appendix A). All energy absorbed beyond the SLED can





be ascribed to internal absorption. Hence, for a layer ranging from top $z_{\text{top}}$ to bottom $z_{\text{bot}}$ that is completely above the SLED, i.e., $z_{\text{bottom}} \leq z_{\text{sled}}$, and by using Eq. (3), the absorbed internal energy $E_{\text{internal}}$ is given by the dissipated flux:

$$
\begin{aligned}
E_{\text{internal}} &= - \int_{z_{\text{top}}}^{z_{\text{bot}}} \frac{z}{z_{\text{sled}}} \frac{\mathrm{d}F(z)}{\mathrm{d}z} dz, \\
&= \frac{F_{\text{top}}}{z_{\text{sled}}} \left( (z_{\text{top}} + \tau) - (z_{\text{bot}} + \tau) \, \mathrm{e}^{-(z_{\text{bot}} - z_{\text{top}})/\tau} \right).
\end{aligned}
\tag{4}
$$

The absorbed energy that contributes to the SEB is the difference between the total energy absorbed in this layer and $E_{\text{internal}}$. For the layer in which the SLED is located, Eq. (4) is evaluated to $z_{\text{sled}}$; all energy below SLED is counted as internal energy absorption.

Figure 1 illustrates $F(z)$, $\mathrm{d}F(z)/\mathrm{d}z$ and the energy contributing to $E_{\text{SEB}}$ and $E_{\text{internal}}$ as a function of depth for a typical fresh snow (Fig. 1a) and ice layer (Fig. 1b) for Rp3's spectral band 6 (778-1242 nm) and 4 (442-625 nm), respectively. As
energy absorption depends on $F(z)$ and $\tau$, which in turn depends strongly on wavelength and snow structure (Meirold-Mautner and Lehning, 2004; Ackermann et al., 2006; Warren et al., 2006; He et al., 2017, 2018; He and Flanner, 2020; Cooper et al., 2020), significantly more energy is absorbed within the SLED for fresh snow and near-IR radiation, as $\tau$ is small for this case, than for ice and visible light, for which $\tau$ is large, resulting in more internal energy absorption close to surface and a larger $E_{\text{SEB}}$.

## 2.4 RACMO2 simulations

In this study, we test Rp3 on a 11 km grid for the GrIS and its immediate surroundings between 2006 and 2015, using September 2000 to December 2005 as spinup. At the lateral boundaries, Rp3 is forced with ERA-Interim data (Dee et al., 2011). The only impurity type considered is soot, which is homogeneously distributed for all snow and firn layers. The impact of soot is known to be underestimated, so a fixed concentration of 5 ng g$^{-1}$ is used, which is higher than the typical 3 ng g$^{-1}$ soot concentration
of the interior (Chylek et al., 1992; Doherty et al., 2010; Dang et al., 2015). A spatially variable soot concentration is used for bare ice such that the Rp3 bare ice albedos match the MODIS observations for clear-sky conditions and typical solar zenith angle (Van Dalum et al., 2020).

The firn-column, i.e, snow and ice density, thickness, temperature, grain radius and water concentration, is initialized for all active layers with output of Rp2 (Noël et al., 2018b), and all grains are spherically shaped. Bare ice is identified if the
continuous set of layers, counted from bottom up, have a density larger or equal to 899 kg m$^{-3}$.

In order to highlight the impact of model changes, we investigate the sensitivity of various parameters by comparing the results with a run of Rp3 that is without internal energy absorption (Rp3 WIE). Rp3 WIE also covers the same period of analysis as Rp3 and Rp2. Furthermore, we investigate the sensitivity of the numerical choices in the implementation of internal heating.



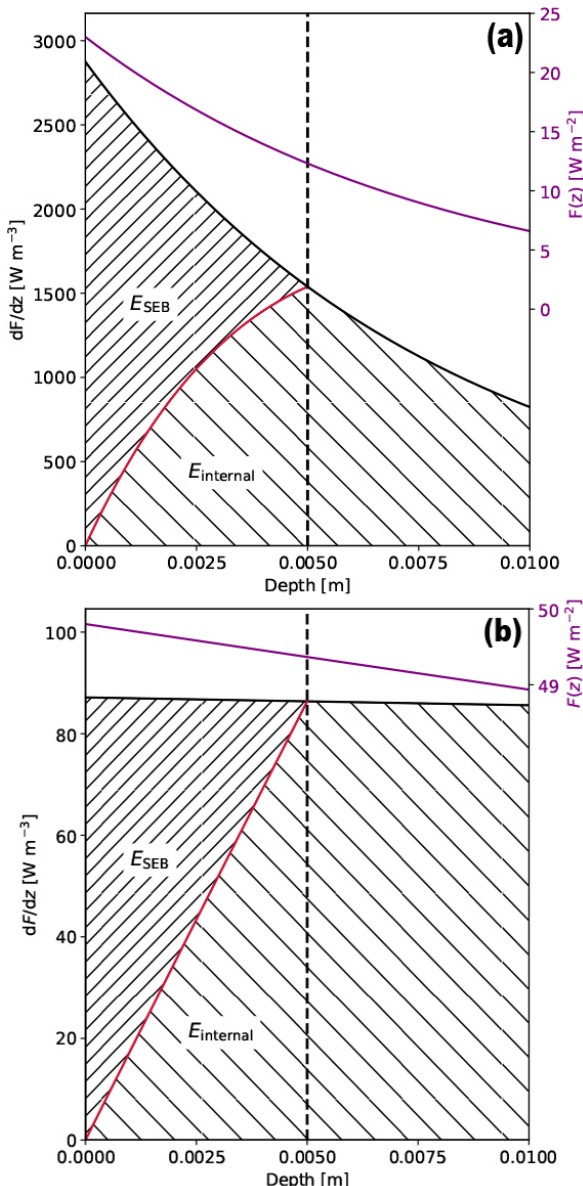

**Figure 1.** Net downward shortwave radiation $F(z)$ (purple), $dF(z)/dz$, absorbed energy contributing to the SEB ($E_{\mathrm{SEB}}$), indicated by the area above the red curve with dense hatches, and internal energy absorption ($E_{\mathrm{internal}}$), indicated by the area under the red curve with sparse hatches, as a function of depth for (**a**) a fresh snow layer and Rp3's spectral band 6 (778-1242 nm) and (**b**) an ice layer for band 4 (442-625 nm). The vertical dashed line shows the 5 mm skin layer equilibration depth (SLED). Here, we assume typical net downward shortwave radiation at the surface and $\tau = 0.008$ m for (**a**) (Meirold-Mautner and Lehning, 2004) and $\tau = 0.56$ m for (**b**) (Cooper et al., 2020). All absorbed energy beyond $z_{\mathrm{sled}}$ is internal.





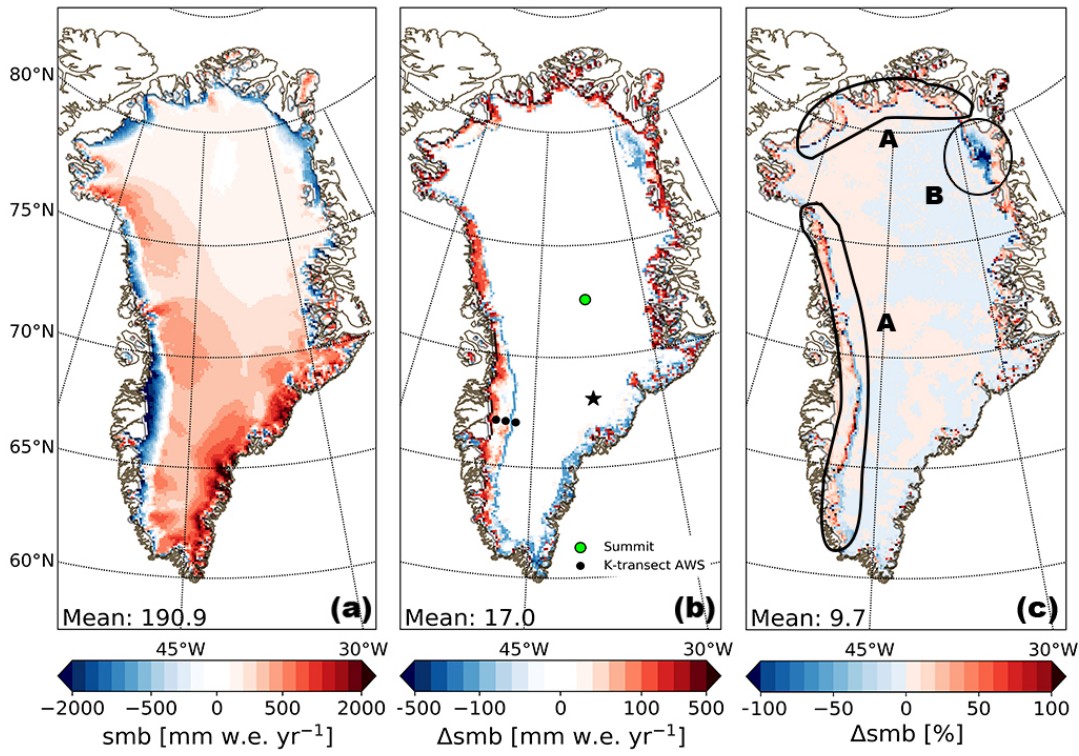

**Figure 2.** (**a**) Average surface mass balance between 2006 and 2015 for Rp3, with a linear color scale with a stepsize of 100 mm w.e. yr$^{-1}$ between -500 and 500 mm w.e. yr$^{-1}$ and 500 mm w.e. yr$^{-1}$ elsewhere. (**b**) SMB difference (Rp3 -Rp2). A linear color scale with a step size of 100 mm w.e. yr$^{-1}$ between -500 and -100 and between 100 and 500 mm w.e. yr$^{-1}$ is used and 20 mm w.e. yr$^{-1}$ elsewhere. The dots in the southwest are the locations of the K-transect AWS stations, from left to right: S6, S9 and S10. KAN-U and KAN-M of the PROMICE data set are located close to S10 and between S6 and S9, respectively, but are not shown separately. Subsurface temperature profiles are available for Summit. The star indicates a location that will be discussed in Sect. 6.1. (**c**) The relative difference between Rp3 and Rp2. Patterns **A** and **B** are discussed in the text.

## 2.5 In-situ observations

Three type of measurements are used in this study: stake measurements to determine the SMB, AWS observations to evaluate the SEB, 2-m temperature ($T_{2m}$) and 10-m wind speed ($v_{10m}$), and subsurface temperature observations. The SMB observational data set in the GrIS ablation zone of Machguth et al. (2016) consists, among others, of data along the Kangerlussuaq transect (K-transect, Smeets et al., 2018) and data of the Programme for Monitoring of the Greenland Ice Sheet (PROMICE, Van As et al., 2011). The K-transect is located perpendicular to the ice margin at approximately 67° N in southwest Greenland and includes both ablation and accumulation sites. PROMICE measurement sites are positioned around Greenland, but are mostly located in the ablation zone. AWS data of both the K-transect and PROMICE are used to evaluate the SEB. Deriving energy fluxes from AWS data is not trivial, and requires assumptions regarding surface roughness and sensor tilt corrections.





For the K-transect, this is discussed in more detail in Smeets et al. (2018). Furthermore, subsurface snow temperature profiles
from Summit are used. These temperature profiles have been measured during the summer of 2007 as part of the Summit
Radiation Experiment (SURE '07) at a depth of 0.02 m up to 0.10 m using thermocouples and at a depth of 0.20, 0.30, 0.50,
0.75 and 1.00 m using thermistor strings (Kuipers Munneke et al., 2009).

The majority of observations are performed outside the time window of the model run or within two grid points of the
ice-sheet margin; these are dismissed, as the ice-sheet margins are not properly captured at the 11 km resolution used for
these runs. For example, the modeled surface elevation above sea level for the PROMICE stations NUK-U and THU-U, both
located close to the ice margin within two grid points on the RACMO2 grid, show an undesirably large elevation difference
with observations: 1069 and 665 m in Rp3 versus 1120 and 760 m observed, respectively. This limits the data set to S6, S7,
S8, S9 and S10 for the K-transect, to KAN-M and KAN-U for PROMICE and to two SMB measurement locations slightly to
the northwest of S9.

For the evaluation of SMB and SEB, the two closest grid points to an observational site are selected in RACMO2 and linearly
interpolated between them. Interpolation between two grid points is, however, not desirable for the analysis of subsurface
processes, as the size and depth of snow layers in RACMO2 can alter considerably between two neighboring points. Instead,
we use the nearest model grid point for subsurface processes.

## 3  Surface mass balance

### 3.1  Comparison with RACMO2.3p2

Figure 2a shows the average 2006-2015 SMB for Rp3. The absolute and relative difference with Rp2 (Rp3 - Rp2) is shown in
Fig. 2b and c, respectively. The SMB changes are typically significant with respect to the inter-annual variability if they exceed
20 mm w.e. yr$^{-1}$. The difference between Rp3 and Rp2 for snow melt, refreezing and runoff are shown in Fig. 3. Changes in
modeled precipitation, sublimation and drifting snow erosion are not shown, as these processes are not significantly affected
by the changes implemented in Rp3 and the differences are on average below 1.1 mm w.e. yr$^{-1}$.

Integrated over the ice sheet, the SMB increases by 17.0 mm w.e. yr$^{-1}$, or 9.8%. In the interior, significant differences are
observed only in south Greenland, with a considerable increase of snow melt (Fig. 3a). As radiation penetration now allows
internal heating, it consequently raises subsurface temperatures and increases melt. Because all snow melt refreezes in the
snow pack, these changes do not affect the local SMB (Fig. 3b and c). Refreezing, however, changes the structure of the snow
pack, inducing more energy absorption and higher snow temperatures, leading to more melt.

The outer rim of the ice sheet, except in the southeast, is characterized by a strong SMB increase. In Rp2 at 11 km, the bare
ice albedo of the majority of the outermost glaciated points is 0.30 due to contamination with tundra albedo, causing too much
melt and runoff. This artifact is mitigated for higher resolutions and is solved in Rp3 (Van Dalum et al., 2020), lowering the
snow melt and runoff and increasing the SMB. In the southeast close to the ice-sheet margin, this issue is absent, as at this
low resolution the number of grid points for which bare ice reaches the surface is limited. Snow melt is slightly higher and
refreezing is similar to Rp2 for this region, resulting in more runoff. This is discussed in more detail in Sect. 5.

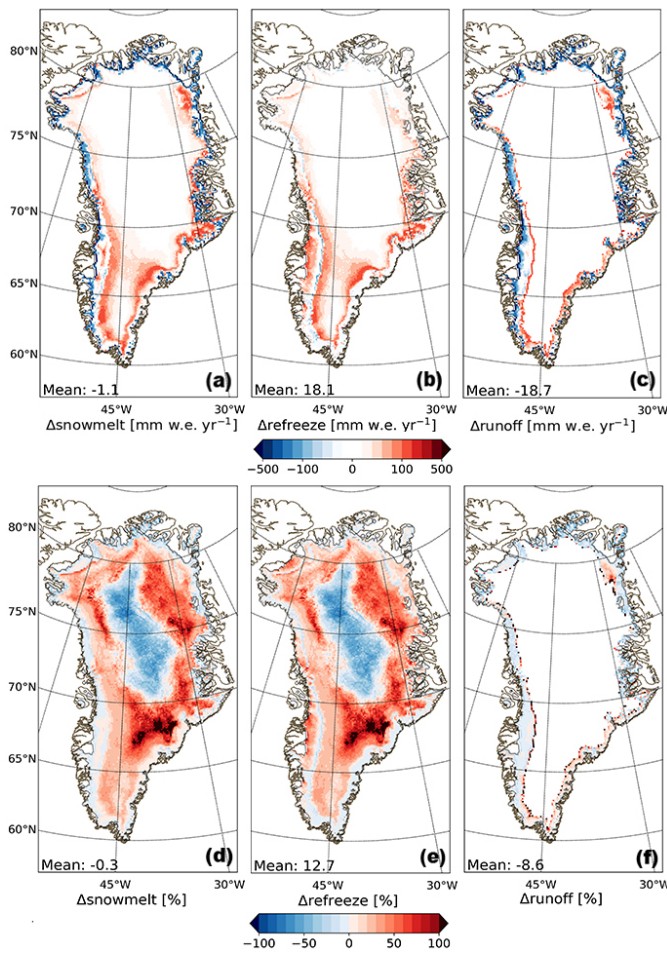

**Figure 3.** Average difference (Rp3 - Rp2) for 2006-2015 for (**a**) snow melt, (**b**) refreezing and (**c**) runoff in mm w.e. yr$^{-1}$, with the same color scale as in Fig. 2b, and average relative difference for (**d**) snow melt, (**e**) refreezing and (**f**) runoff.

The lower ablation area of the GrIS in the southwest experiences a strong increase in refreezing (Fig. 3b), while the SMB difference is limited. Refreezing is enhanced by the introduction of radiation penetration. In Rp3, subsurface melting of ice creates pore space (Van Dalum et al., 2020, Sect. 2.1.1) and consequently increases the melt water retention capacity. In Rp2, no

retention capacity remains once bare ice reaches the surface, limiting refreezing in the shallow winter snowpack. Additionally, some regions in the southwest show a small increase in snow melt, which is induced by, on average, a slightly lower albedo in Rp3 (Van Dalum et al., 2020). As a result, the increase in snow melt and refreezing balance out, leading to a runoff similar to Rp2 (Fig. 3) and little change in SMB (Fig. 2b).

In Rp3, snow melt has increased in the percolation zone and most of the additional melt water refreezes locally. Close to the

equilibrium line, however, the melt water buffering capacity is exceeded due to the addition of melt water and runoff occurs,

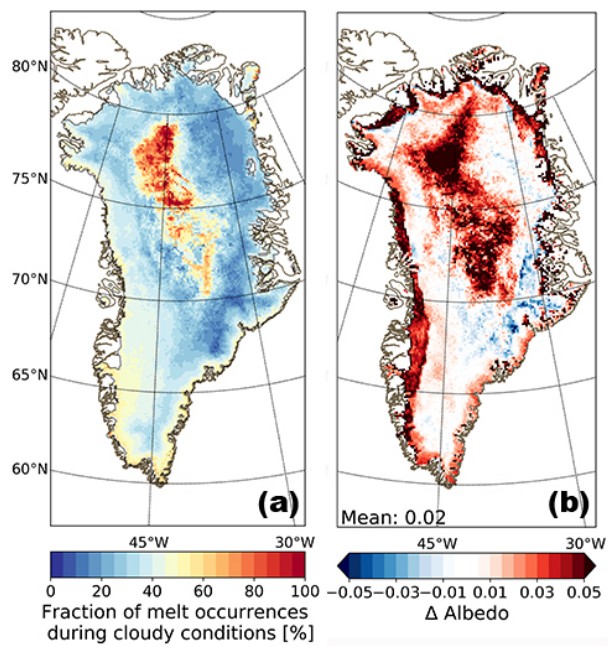

**Figure 4. (a)** Fraction of melt occurrences during cloudy conditions for Rp3 and **(b)** the albedo difference (Rp3 - Rp2) during melt events and cloudy conditions. A minimum vertically integrated cloud content of 0.05 kg m$^{-2}$ is required to be considered cloudy and a minimum melt rate of 250 mm w.e. year$^{-1}$ is required to be considered a melt event. All data are averaged between 2006 and 2015.

lowering the SMB (pattern **A** in Fig. 2c). Similarly, in the upper ablation zone, where the winter snow cover lasts long during summer, enhanced melt leads to a small reduction of refreezing, as the refreezing capacity is consumed faster.

The northeast shows a large area with reduced SMB (pattern **B** in Fig. 2c). This region is characterized by dry conditions, resulting in a thin winter snow layer that melts away quickly in spring, exposing bare ice. Due to the low albedo of bare ice

and radiation penetration, the ice temperature is raised, leading to melt in multiple model layers reaching greater depths and prolonging the melt season (discussed in more detail in Sect. 6.1), consequently enhancing melt and runoff (Fig. 3a, c, d and f). Furthermore, it usually takes several weeks after the ablation season ends for fresh snow layers to grow to a thickness of over 10 cm. During this time, some radiation can still reach bare ice layers, lowering the albedo and inducing more energy absorption.

To summarize, averaged over the ice sheet, snow melt has barely changed, i.e., -1.1 mm w.e. yr$^{-1}$ (-0.33%), compensating the increase of melt in south Greenland with the reduced melt at the margins. The runoff, on the other hand, is lowered by -18.7 mm w.e. yr$^{-1}$ (-8.6%) while refreezing has increased by 18.1 mm w.e. yr$^{-1}$ (12.7%). By percentage, snow melt and refreezing have increased considerably in the interior of the ice sheet, except for a large area in the center (Fig. 3d and e). This is discussed in more detail in Sect. 3.2. No runoff is modeled in the interior (Fig. 3c and f), as melt water refreezes locally.

Around the K-transect, the relatively large increase of refreezing consequently lowers the modeled runoff. A relatively strong runoff increase is modeled in **A** and **B**.

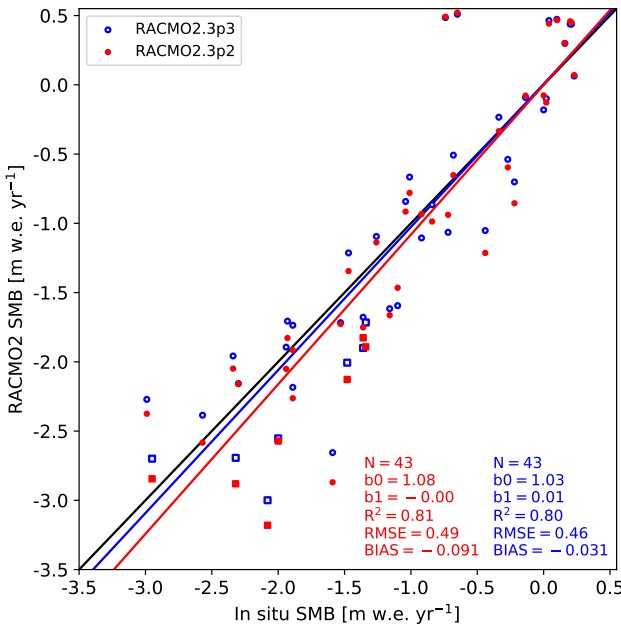

**Figure 5.** Surface mass balance along the K-transect in Rp3 (blue) and Rp2 (red), compared to in-situ stake observations (m w.e. yr$^{-1}$). Measurements are taken between 2006 and 2015, with each dot representing one year. Only observations at least 2 grid points away from the ice margin in the RACMO2 grid are selected for evaluation (locations are shown in Fig. 2b). The black line is the 1-on-1 line and the red and blue line show the linear regression of the data, with b0 the slope and b1 the intercept. Number of records (N), correlation coefficient ($R^2$) and root-mean-square error (RMSE, in m w.e. yr$^{-1}$) are also displayed. Results for S6 are shown with squares.

## 3.2 Cloud cover and melt

For a large area of central GrIS, snow melt and refreezing have decreased considerably by percentage (Fig. 3d and e). Melt events in this region, although rare, are almost always associated with above-average cloud cover (Fig. 4a), reducing the amount

of SW radiation that reaches the surface. In general, the albedo of Rp3 is lower than Rp2 for this region (Van Dalum et al., 2020), but for cloudy conditions and during melt events, the albedo of Rp3 is considerably higher (Fig. 4b), leading to less energy absorbed in the snow pack and subsequently less melt and refreezing. This albedo increase is mostly caused by the changing spectral distribution of irradiance during cloudy conditions, as relatively more irradiance is ultraviolet (UV) and visible light, for which the albedo is high, and relatively less is IR radiation, for which the albedo is low (Van Dalum et al.,

2019). A large fraction of UV and visible radiation is absorbed internally, heating subsurface layers that can potentially increase melt. This is discussed in more detail in Sect. 5.2. This effect, however, is not enough to mitigate the reduced irradiance and higher albedo and therefore less snow melt is modeled. Note that the total amount of snow melt modeled for this region in central GrIS is on average very small (2 mm w.e. yr$^{-1}$) compared to ablation areas (e.g., 2500 mm w.e. yr$^{-1}$ at S6).



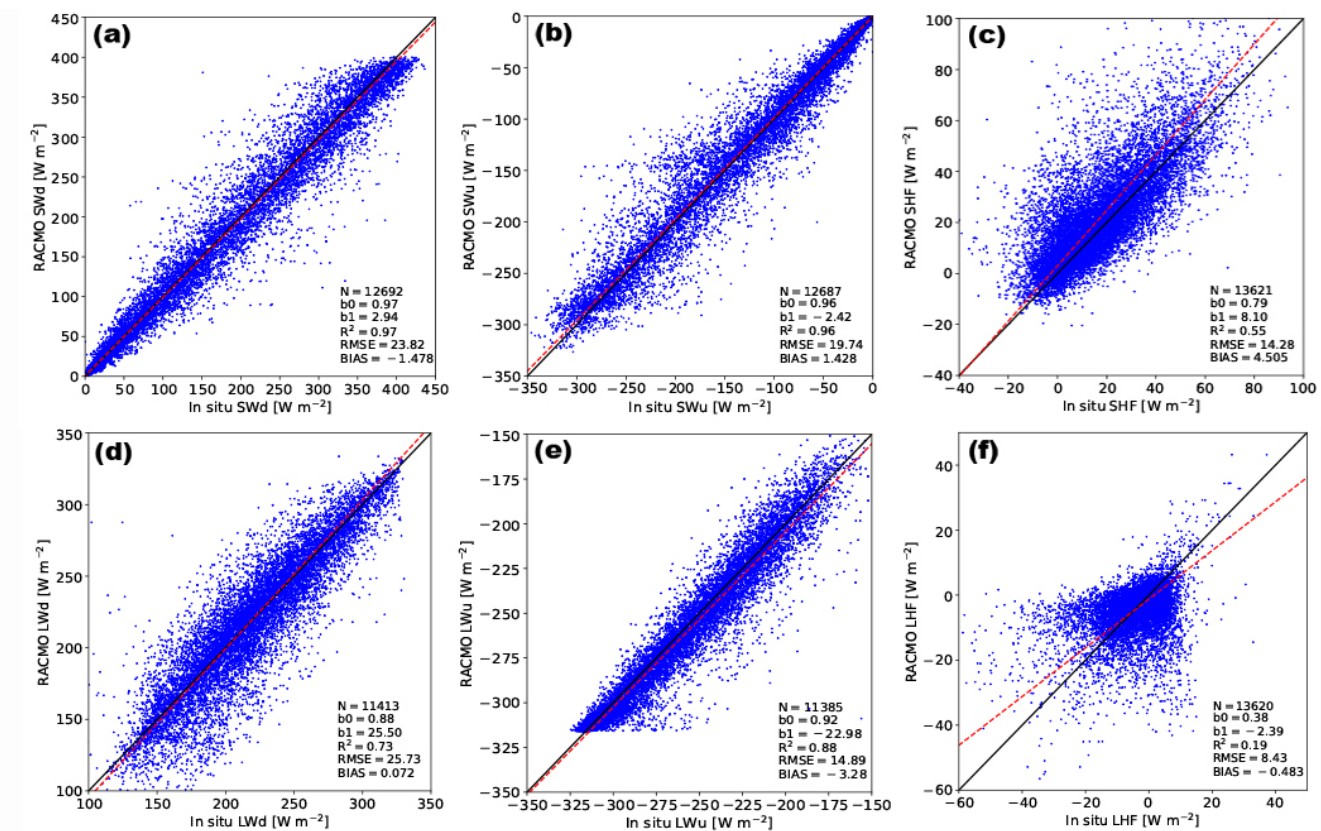

**Figure 6.** Daily-averaged Rp3 SEB components between 2006 and 2015 compared to in-situ observations including S6, S9 and S10 of the K-transect and KAN-U and KAN-M of PROMICE for **(a)** downward shortwave radiation (SWd), **(b)** upward shortwave radiation (SWu), **(c)** sensible heat flux (SHF), **(d)** downward longwave radiation (LWd), **(e)** upward longwave radiation (LWu) and **(f)** latent heat flux (LHF). The black line is the 1-on-1 line and the red line shows the orthogonal total least squares regression of the data, with b0 the slope and b1 the intercept. Number of records (N), correlation coefficient ($R^2$), root-mean-square error (RMSE, in W m$^{-2}$) and bias are also displayed. In all panels, positive values imply an energy flux directed towards the surface.

### 3.3 SMB Observations

Figure 5 shows the SMB of Rp3 and Rp2 with respect to various SMB observations. The correlation coefficient, root-mean-square error (RMSE) and bias with respect to observations are also shown. A slightly higher SMB is modeled for Rp3 compared to Rp2, reducing the bias with observations. The differences with observations are particularly relevant for the measurement sites in the lower ablation zone (S6, squares in Fig. 5) and are caused by increased refreezing in Rp3 (Fig. 3b). Note that the spread is considerably higher for the ablation zone, illustrating that the large temporal variability is not always captured

properly with this resolution in RACMO2. Increasing the resolution might improve the modeled estimates for the locations close to the ice margin.



**Table 1.** Statistics of the daily-averaged Rp2 and Rp3 SEB components, atmospheric 2-m temperature ($T_{2m}$) and 10-m wind speed ($V_{10m}$) compared to in-situ observations between 2006 and 2015. Data include S6, S9 and S10 (S10 is not available for $T_{2m}$ and $V_{10m}$) of the K-transect and KAN-U and KAN-M of PROMICE. The correlation coefficient ($R^2$), root-mean-square error (RMSE) and bias are shown.

| Variable | Unit | Rp2 | | | Rp3 | | |
|---|---|---|---|---|---|---|---|
| | | $R^2$ | bias | RMSE | $R^2$ | bias | RMSE |
| LWd | W m$^{-2}$ | 0.72 | 0.1 | 25.9 | 0.73 | 0.1 | 25.7 |
| LWu | W m$^{-2}$ | 0.88 | -2.6 | 14.8 | 0.88 | -3.3 | 14.9 |
| SWd | W m$^{-2}$ | 0.96 | -4.2 | 26.2 | 0.97 | -1.5 | 23.8 |
| SWu | W m$^{-2}$ | 0.94 | 3.0 | 23.0 | 0.96 | 1.4 | 19.7 |
| SHF | W m$^{-2}$ | 0.60 | 4.7 | 13.5 | 0.55 | 4.5 | 14.3 |
| LHF | W m$^{-2}$ | 0.20 | -0.4 | 8.3 | 0.19 | -0.5 | 8.4 |
| $T_{2m}$ | °C | 0.95 | 1.4 | 2.7 | 0.95 | 1.6 | 2.8 |
| $V_{10m}$ | m s$^{-1}$ | 0.76 | 0.3 | 1.7 | 0.75 | 0.3 | 1.8 |

## 4   Surface energy budget

Figure 6 shows that the SEB components of Rp3 correlate generally well with observations of S6, S9 and S10 of the K-transect and KAN-U and KAN-M of PROMICE. The downward and upward longwave radiative flux (LWd and LWu) show a correlation coefficient ($R^2$) of 0.73 and 0.88, respectively and the performance of Rp3 is similar to Rp2 (Table 1). The small bias of LWd and shortwave downward radiative flux (SWd) shows that the influence of cloud cover is captured well. There is a tendency for Rp3 to underestimate LWd during cold and dry, cloud-free winter (small SWd) conditions, resulting in underestimated LWu for the same cold days.

SWd and upward shortwave radiative flux (SWu) are both in good agreement with observations. The $R^2$, bias and RMSE of these fluxes have improved in Rp3 compared to Rp2 (Table 1), which confirms that the albedo has improved, in agreement with Van Dalum et al. (2020).

Both the sensible and latent heat flux (SHF and LHF, respectively), show a considerable spread (Fig. 6c and f). While the LHF is small and contributes little to the SEB, the SHF is important to model melt events properly. The difference with observations for the 2-m temperature ($T_{2m}$) 2-m and 10-m wind speed ($V_{10m}$) (Table 1 and Fig. B1), however, are relatively small, and can therefore not explain the large spread of the turbulent fluxes. As is discussed in Noël et al. (2018b), the bias in SHF can be attributed to an inaccurate representation of the roughness length, especially for bare-ice conditions.

Figure 7 shows the climatology of the SEB components of in-situ, Rp2 and Rp3 data for S6, S9 and S10. The Rp3 and Rp2 radiative and turbulent fluxes are generally similar, and agree with in-situ observations. Some differences with observations are, however, still observed. The net absorbed longwave radiation (LWn), defined as LWd - LWu, is significantly lower during winter for both Rp2 and Rp3, which is mostly due to the temperature error induced by an overestimation of SHF and partly


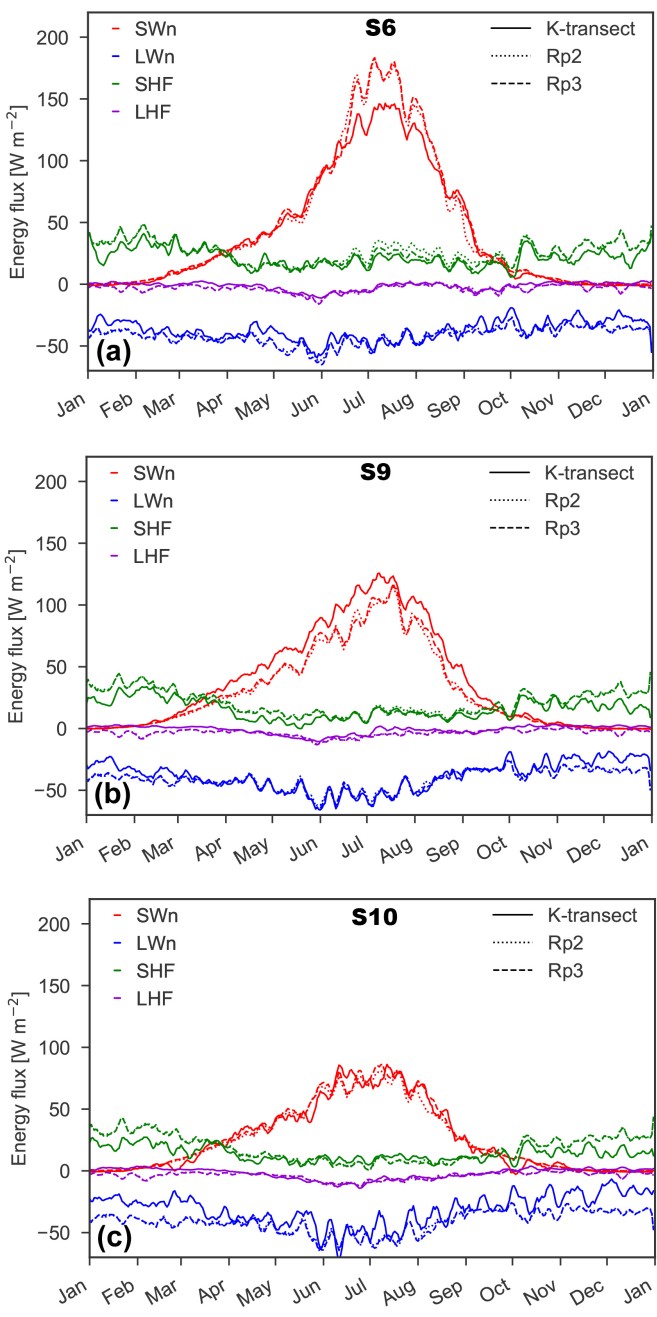

**Figure 7.** 2006-2015 averages of the 5-day running mean SEB components for the K-transect for in-situ observations, Rp2 and Rp3 at **(a)**
S6 and **(b)** S9, and for 2010-2015 at **(c)** S10.

due to a small negative bias of LWd. No changes have been made to the cloud and SHF parameterizations in Rp3, so similar





results as Rp2 are therefore expected (Noël et al., 2018b). Improving the representation of surface roughness may also improve the results. During summer, only the net absorbed shortwave radiation (SWn), defined as SWd - SWu, shows considerable discrepancies for S6 and S9. The introduction of a new radiation penetration and snow albedo scheme reduces the differences with observations for Rp3 compared to Rp2. This is especially noticeable at the onset of the accumulation season at S6 when a thin fresh snow layer forms on top of bare ice. As Rp2 does not include radiation penetration, the ice layers are colder and the melt season ends earlier, allowing the early formation of a fresh snow layer, raising the albedo. The albedo differences are relatively small for S10, as this location is characterized by a thick firn layer. The evaluation of the snow albedo scheme of Rp3 is discussed in more detail in Van Dalum et al. (2020).

## 5 Subsurface energy absorption

### 5.1 K-transect

Internal energy absorption of shortwave radiation is an important addition in Rp3. We illustrate the absorption of solar radiation and grain radius as a function of depth for the K-transect for 2012 in Fig. 8. S6 is situated in the ablation zone and this dry region is characterized by a relatively thin fresh snow pack after winter. During June, the ice horizon quickly moves up exposing bare ice until September. As S6 is situated in the so-called 'dark zone' (Van de Wal and Oerlemans, 1994; Wientjes et al., 2011), the exposed bare ice has a high impurity concentration and a large grain radius, leading to extensive internal energy absorption up to 15 cm deep. During the accumulation season, internal energy absorption is limited, except during sporadic melt events when the surface grain radius grows rapidly, allowing radiation to penetrate to deeper layers.

At S10, situated in the accumulation zone, higher winter accumulation provides a thicker snow pack at the start of the melt season than at S6. Consequently, the whole snow column only melts away during extreme melt events like in the summer of 2012 and 2019, in which bare ice surfaces for a few weeks in August (Fig. 8b). In late June and July, firn layers that are characterized by a large grain radius reach the surface and induce significant absorption of solar radiation, although less than for bare-ice conditions. The absorption, however, is limited to the upper 5 cm. Note that the formation of a thin fresh snow layer in early July considerably diminishes internal energy absorption.

### 5.2 Distribution of energy

Since a significant fraction of the shortwave radiation is absorbed in the upper few millimeters of the snow column, not all incoming radiation contributes to internal heating (Sect. 2.3). The fraction of shortwave radiation absorbed that directly contributes to the SEB depends on wavelength and snow conditions, which is illustrated in Fig. 9. This figure shows, as an example, the absorption of energy for a summer day for a grid point at Summit in central Greenland (left bars) and S6 (hatched right bars) for the first twelve spectral bands. Band 13 and 14 are not shown, as all energy for these bands is added to the SEB. Blue shows the energy absorbed internally and orange the energy that contributes to the SEB, with numbers above the bars indicating the percentage of surface radiation that is absorbed.





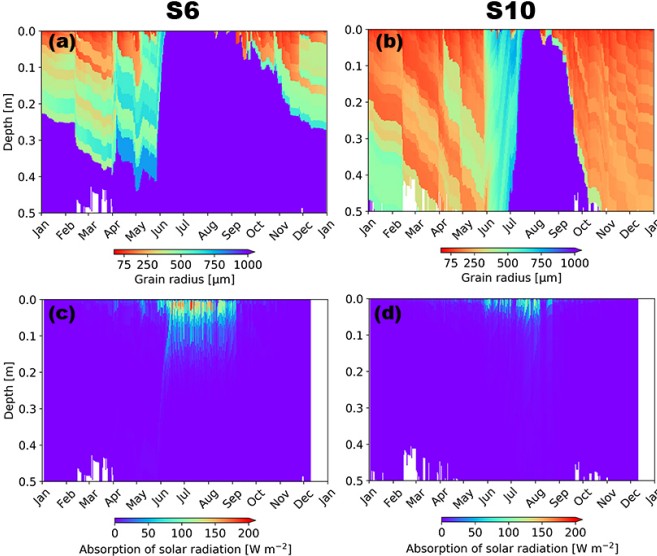

**Figure 8. (a)**, **(b)** Grain radius and **(c)**, **(d)** absorption of solar radiation as a function of time and depth for the 20 upper most snow layers for S6 and S10, respectively, for local solar noon (15:00 UTC) in 2012. Bare ice is indicated by a grain radius of 1000 $\mu$m and white bars indicate layers beyond the 20 upper snow layers.

At Summit, almost all radiation is absorbed internally for visible light (bands 1-5), but the amount of radiation that remains in the snow pack for these bands is limited. For IR radiation (bands 6-12), considerably more absorption takes place. The
attenuation length in snow is short for IR radiation (He and Flanner, 2020), leading to a strong absorption within the SLED, increasing the fraction of energy that contributes to the SEB.

S6 has bare ice at the surface for the selected day, and considerably more radiation is absorbed in the visible light bands due to a high concentration of soot. Most notably, is the large increase of energy absorbed in band 6. This is a wide band with a considerable amount of energy available. It is especially sensitive to albedo drops induced by an increase of grain radius and
density, e.g., when bare ice reaches the surface (Van Dalum et al., 2019).

For bands 7 to 12, almost all incoming radiation is absorbed at S6 due to the large grain radius and density. For these bands a larger fraction of energy contributes to the SEB at S6 compared to Summit. As the grain size at S6 has increased more strongly than the density when compared to the snow pack at Summit, the number of grains in the SLED is therefore lower. In other words, the snow pack at S6 misses a scattering fresh snow top layer, hence increasing the chance of near-IR radiation to
penetrate through the SLED without being absorbed. This effect is less pronounced for visible light, as the SLED is so small that visible light travels in and out virtually without loss, especially if the soot concentration is as low as that at Summit.





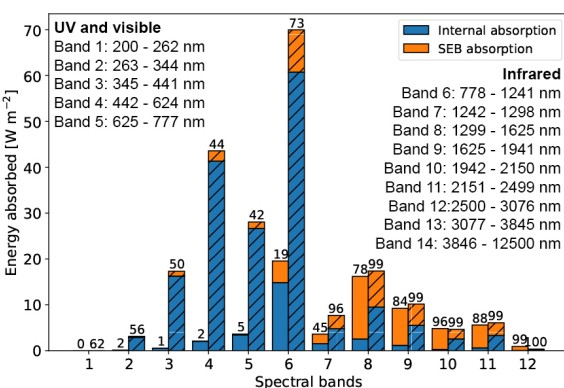

**Figure 9.** Energy absorbed in the snow pack during a clear-sky summer day (10 July 2007) at Summit in central Greenland (left bars) and S6 (right hatched bars) for the first twelve spectral bands of Rp3. Blue indicates the fraction of energy that is absorbed internally, orange the fraction that contributes directly to the SEB. Numbers above the bars show the percentage of radiation absorbed with respect to incoming radiation at the surface. In bands 13 and 14 (not shown), absorption at the surface is 100%.

## 6 Sensitivity experiments

In this section, we discuss the impact of internal energy absorption on the subsurface temperature for various locations in winter and summer. Furthermore, we investigate the 10-m snow temperature ($T_{10m}$) and the impact of internal energy absorption on the SMB by comparing the results with Rp3 without internal energy absorption (WIE).

### 6.1 Subsurface temperature

First, we analyze the impact of internal energy absorption on subsurface snow temperatures. Figure 10 shows the subsurface temperature profile for Rp2, Rp3 and Rp3 WIE for S6 and Summit for a typical winter day and summer day, and for an extraordinary warm summer day for S6 and a grid point in southeast Greenland (locations are shown in Fig. 2b). As the modeled temperature is not interpolated between layer mid-points, the stepwise temperature profile indicates individual model layers. Rp3 generally has thinner snow layers, leading to a higher vertical resolution, which is especially relevant close to the surface. As Fig. 10 only shows the upper 20 snow layers, the temperature curve of Rp3 therefore does not reach as deep as Rp2.





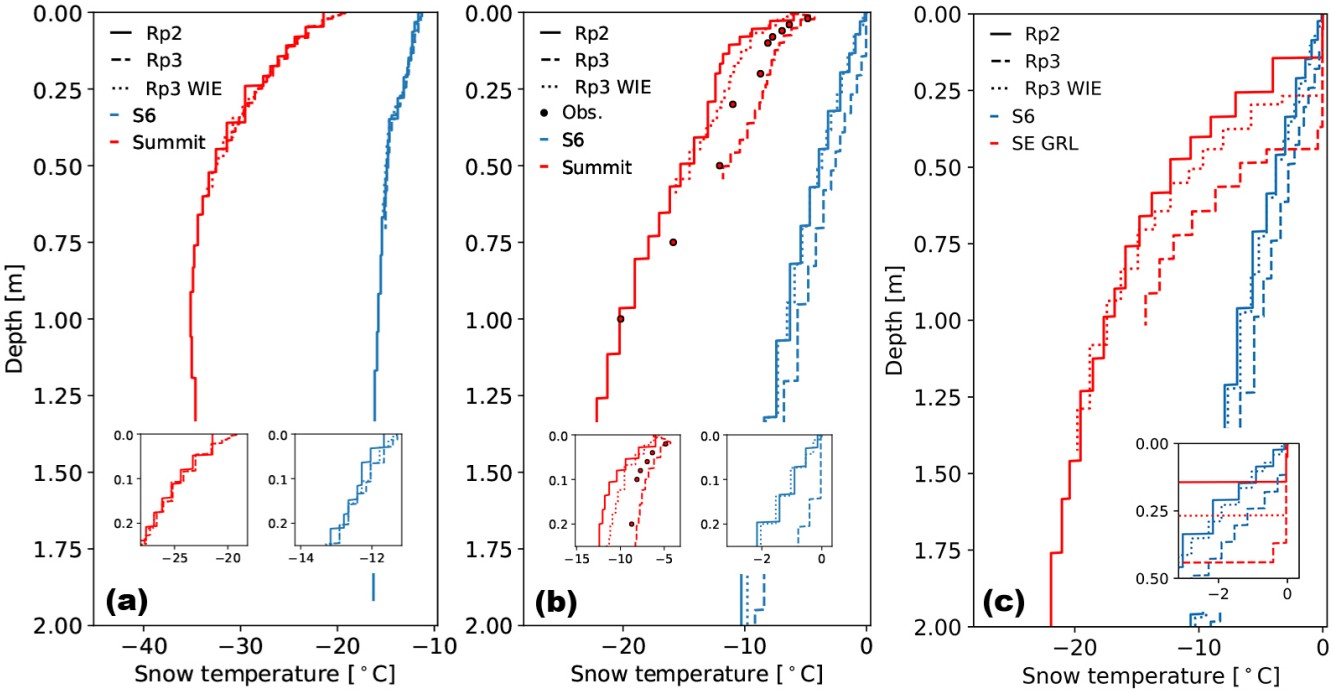

**Figure 10.** Subsurface temperature profile for the 20 upper snow layers of Rp2, Rp3, Rp3 without internal energy absorption (WIE) and observations (Obs.) for S6 and Summit for **(a)** a winter day (29 January 2007) and **(b)** a summer day (10 July 2007) at 15:00 UTC. No observations are available for Summit on this winter day. **(c)** Subsurface temperature profile for a location in the accumulation zone of southeast Greenland (SE GRL, star in Fig. 2b) and S6 during a melt event (22 June 2012 15:00 UTC). The insets show the results of the upper layers in more detail.

During winter (Fig. 10a), the temperature profiles are similar for both S6 and Summit. Internal heating is limited, as only a
small amount of radiation reaches the surface and both sites are covered with fine-grained fresh snow layers. Only a significant change is observed in the upper first centimeters, with thinner snow layers and higher temperatures in Rp3.

During summer (Fig. 10b), internal energy absorption has a clear impact on subsurface temperatures. For S6, melting occurs in multiple layers reaching to a depth of about 14 cm, which coincides approximately with depths that absorption of solar radiation is still relevant (Fig. 8c). With the addition of internal heating, the temperature of deeper layers is raised as internal
heat cannot dissipate as easily as on the surface. Higher subsurface snow temperatures make the snowpack more susceptible to internal melt, thus increasing the vertical melt extent. Note that all melting layers are still connected to the surface, effectively reducing it to a single melt layer. The impact of internal heating is also illustrated by the similar temperature profile of Rp3 WIE with respect to Rp2.

For Summit, much colder conditions prevail during summer. Compared to Rp2, the subsurface temperatures of Rp3 are
considerably higher (up to 5 °C) and match the observations better, but are slightly overestimated in the upper 10 centimeters.





Rp3 WIE shows lower temperatures similar to Rp2, leading to a cold bias with the observations. The temperature decline with depth, however, is more gradual for Rp3 WIE compared to Rp2. Adding internal energy absorption considerably increases the ability of RACMO2 to reproduce realistic subsurface temperature profiles for Summit.

Figure 10c shows the subsurface temperature profile during a melt event in the accumulation zone of southeast Greenland

(SE GRL, star in Fig. 2b) and for S6. The figure illustrates that melting point is reached up to a considerably greater depth for the accumulation point in SE GRL than S6 (0.36 m and 0.18 m depth, respectively). This can be understood by surface melt percolating through snow to deeper layers, warming the subsurface. In ice at S6, melt water can only percolate downwards through pores that are generated by internal melting. The snow pack in SE GRL, on the other hand, consists of various layers of relatively fresh snow and firn where melt water can easily percolate, similar to the snow pack illustrated for S10 in May

in Fig. 8b. A larger vertical melt extent is therefore expected for this grid point. Moreover, the addition of internal heating enhances subsurface temperatures even more, further increasing the vertical melt extent. As a warmer snowpack is more prone to melting, more melt events are expected to occur in the accumulation zone like in SE GRL. This is in agreement with Fig. 3d and e, which shows a relatively strong increase in snow melt and refreezing around central GrIS for Rp3 compared to Rp2, especially in the southeast where this grid point is located. As this location in SE GRL is too far from the ice margin, no runoff

is modeled and all melt water refreezes locally. Even though the snow melt and consequent refreezing are still small and do not alter the SMB, they do change the snow structure.

We also investigate the sensitivity of the subsurface snow temperature to the the skin layer equilibration depth $z_\mathrm{sled}$, which we introduced in Sect. 2.3 and set to 5 mm using scale analysis. Underestimating $z_\mathrm{sled}$ leads to an overestimation of internal heating, as not enough heat diffuses to the surface within the model time step, trapping too much heat beneath the surface.

Increasing $z_\mathrm{sled}$ only has a limited effect on the subsurface temperature (Appendix Fig. B2). As expected, lowering $z_\mathrm{sled}$ results in too much internal heating and raises the snow temperature, illustrating that an underestimation of $z_\mathrm{sled}$ should be avoided.

### 6.2 Snow temperature at 10 m depth

In the absence of melt and refreezing, the $T_\mathrm{10m}$ is regarded as a good proxy of the mean annual surface temperature. Here, we

analyze whether this assumption is true given the observed impact of radiation penetration on summer subsurface temperatures. Figure 11 shows the averaged yearly mean $T_\mathrm{10m}$ at the end of the simulation for Rp3 (Fig. 11a), the difference between the average skin temperature ($T_\mathrm{skin}$) and $T_\mathrm{10m}$ (Fig. 11b) and the $T_\mathrm{skin} - T_\mathrm{10m}$ difference for Rp3 and Rp3 WIE with respect to Rp2 (Fig. 11c and d, respectively).

In the interior, the difference between $T_\mathrm{skin}$ and $T_\mathrm{10m}$ is small for Rp3, illustrating that $T_\mathrm{10m}$ is indeed a good proxy of the

mean annual surface temperature for this region. For Rp3, $T_\mathrm{skin} - T_\mathrm{10m}$ is comparable to Rp2 (Fig. 11c), for Rp3 WIE, however, it is considerably larger (Fig. 11d). As all solar energy is absorbed at the surface for Rp3 WIE and the albedo is slightly lower compared to Rp2, $T_\mathrm{skin}$ is consequently increased while the energy that reaches 10 m depth is decreased, thus lowering $T_\mathrm{10m}$. In other words, Fig. 11d highlights the importance of radiation penetration.

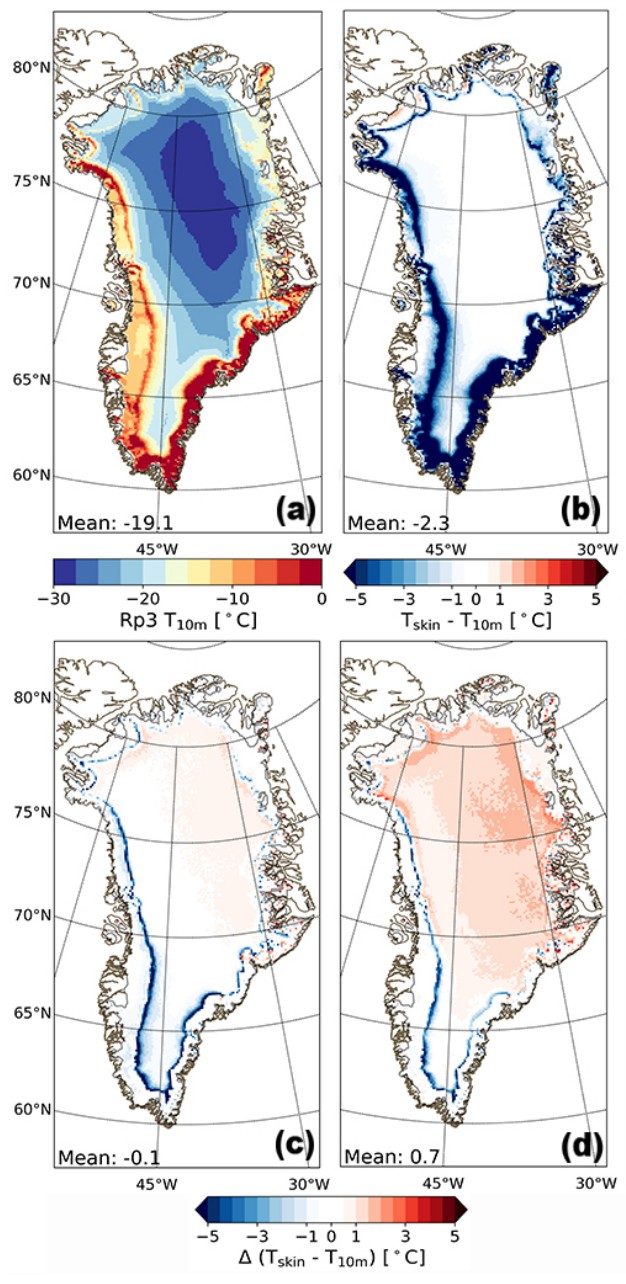

**Figure 11. (a)** Temperature at 10 m depth ($T_{10m}$) at the end of the simulation (31 December 2015) for Rp3; **(b)** Averaged yearly mean (2006-2015) skin temperature ($T_{skin}$) difference with $T_{10m}$ for Rp3, with positive values indicating that $T_{skin}$ is larger than $T_{10m}$; **(c)** Difference between Rp3 and Rp2 for $T_{skin} - T_{10m}$ and **(d)** the difference between Rp3 WIE and Rp2 for $T_{skin} - T_{10m}$. In **(c)** and **(d)**, a positive value indicates that the $T_{skin} - T_{10m}$ has become larger compared to Rp2.





In the percolation zone (**A** of Fig. 2c), the difference between $T_{\mathrm{skin}}$ and $T_{\mathrm{10m}}$ is considerable (Fig. 11b), owing to latent heat

being released upon refreezing. Consequently, $T_{\mathrm{10m}}$ is not representative for the surface temperature in this zone. There are also large differences in southeast Greenland, which is characterized by firn aquifers that keep the deeper snow temperature near the melting point year round.

To conclude, $T_{\mathrm{10m}}$ is a good measure for the surface temperature, except close to the ice-sheet margin in the southeast and the percolation zone. Moreover, subsurface energy absorption reduces $T_{\mathrm{skin}} - T_{\mathrm{10m}}$ in the interior.

## 6.3   SMB without internal energy absorption

In this subsection, we discuss the impact of radiation penetration on the SMB and analyze its components by comparing Rp3 WIE with Rp3. Figure 12 shows the SMB, snow melt and refreezing difference between Rp3 WIE and Rp3. Note that in Rp3 WIE radiation penetration is still used to determine the albedo, but all absorbed shortwave radiation is added to the SEB.

The SMB difference of Fig. 12a is limited in the interior, where melt does not run off. The SMB is lower for Rp3 WIE in
most of the ablation zone (Fig. 12a), in particular in the southwest (area **C** of Fig. 12c), where we observe a relatively small snow melt increase (Fig. 12b). More importantly, however, is that internal radiative heating reduces refreezing in the ablation zone (Fig. 12c), especially in the southwest, but also in the northwest and northeast (area **C**, **D** and **B** of Fig. 2c, respectively), so that runoff increases, reducing the SMB. The SMB difference of Fig. 12a is limited in the interior, where melt does not run off. In the central part of the interior, the melt and refreezing differences are limited, supporting the findings of Sect. 3.2. In a broad
elevation band around the center, less snow melt and refreezing is observed, illustrating the importance of subsurface energy absorption, which prevents the surface from reaching the melting point. In south Greenland, the melt difference is considerably smaller than the difference between Rp3 and Rp2 (Fig. 3d), although still not negligible, illustrating that internal heating is not the only relevant process here. This region is also characterized by an albedo decrease (Van Dalum et al., 2020), resulting in more absorption of shortwave radiation and snow melt, contributing significantly to the observed difference between Rp3 and
Rp2.

## 7   Conclusions

In this study, we evaluated the impact of a new snow and ice albedo and radiative transfer scheme that allows internal heating on the surface mass balance (SMB), surface energy budget (SEB) and firn and snow temperature of the GrIS in the latest version of the regional climate model RACMO2. In this new version Rp3, we also updated the multilayer firn module (Van Dalum et al.,
2020) and developed a method to distribute total absorbed shortwave radiation between the SEB and to internal absorption.

We have run Rp3 for the GrIS for 2006-2015 using ERA-Interim data at the lateral boundaries. The modeled SMB correlates well with ablation stake measurements and the SEB is in good agreement with AWS observations at the K-transect and for a selection of PROMICE stations. Moreover, subsurface radiative energy absorption leads to higher summer snow temperatures and a larger vertical melt extent, improving the agreement with observed snow temperatures at Summit.





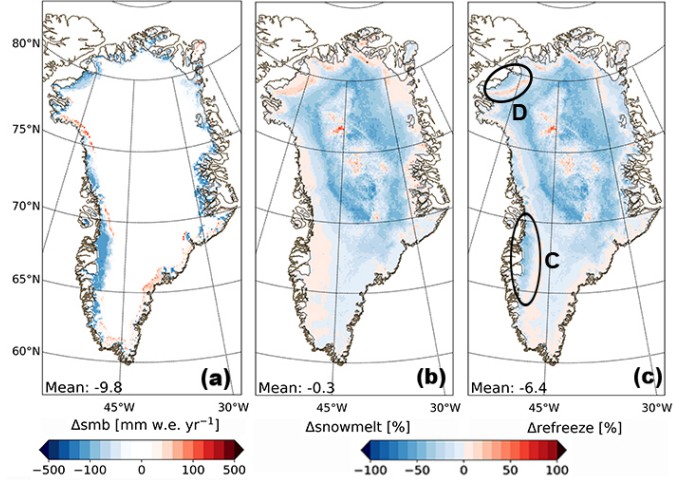

**Figure 12.** 2006-2015 Average difference between Rp3 WIE and Rp3 (Rp3 WIE - Rp3) for **(a)** the SMB in mm w.e. yr$^{-1}$, and by percentage for **(b)** snow melt and **(c)** refreezing. The same color scale is used as in Fig. 2b for **(a)**.

Compared to the previous RACMO2 version Rp2, the SMB difference is small in the interior, as any change in snow melt is balanced by refreezing. In the percolation zone close to the equilibrium line, the SMB is lowered as the melt water buffering capacity is exceeded due to the addition of melt, leading to runoff. In the ablation zone in the southwest, considerably more refreezing occurs due to pore space created in ice by melt induced by subsurface energy absorption. Finally, the SMB is higher at the very margins due to the removal of an artifact in the ice albedo of Rp2, which reduces snow melt. These differences are

small or absent in the wet southeast, where bare ice rarely reaches the surface. For this region, relative differences in snow melt and refreezing are substantial due to subsurface energy absorption and melt water percolation. This increased snow melt and refreezing does not lead to significant SMB changes, but does change the snow structure. Furthermore, subsurface temperature profiles show a more gradual temperature decrease with depth in Rp3 during summer in the interior, and melt reaching deeper layers both in the accumulation and ablation zone.

There is, however, still room for improvement. The resolution used in this study, i.e., 11 km horizontal resolution, is sufficient to resolve the SMB and SEB in the interior of the ice sheet, but is insufficient to resolve the inhomogeneous ablation zone, especially close to the ice margin. For example, bare rock cropping out of the ice, steep slopes or snow patches are usually smaller than the current RACMO2 resolution. More fundamentally, the ice albedo, although significantly improved in Rp3, could be further improved upon by using a dynamic Mie-scattering theory model (Gardner and Sharp, 2010). In addition, a

new impurity scheme that would allow the introduction of more impurity types and varying concentrations for each layer in space and time and would be beneficial.

To conclude, Rp3 is capable to accurately reproduce the SMB, SEB and (sub)surface temperature when compared to in-situ observations. Differences with Rp2 are most pronounced in the ablation zone close to the ice margin, in south Greenland and in the percolation zone. Furthermore, the snow temperature with depth is altered ice-sheet wide due to subsurface energy



absorption. The improvements made in Rp3 are especially relevant in a warming climate where the GrIS surface will typically have lower albedos and more subsurface energy absorption, thus improving RACMO2's capability to make future climate projections. In a subsequent study, we will assess the ability of Rp3 to simulate albedo and melt in Antarctica.

*Author contributions.* CTvD, WJvdB and MRvdB started and decided the scope of this study and analyzed the results. CTvD ran the simulations, implemented the new snow and ice albedo and radiative transfer scheme and led the writing of the manuscript. All authors
contributed to discussions on this work.

*Data availability.* Rp3 (September 2000-2018) and Rp3 WIE (September 2000-2015) monthly-accumulated and monthly-averaged data are available for $T_{2m}$, $T_{10m}$, SMB, SEB and its components at 11 km resolution for Greenland, and can be found here: https://doi.org/10.5281/zenodo.4013856. Rp2 data are available from the authors.

*Competing interests.* The authors declare that they have no conflict of interest.

*Acknowledgements.* We acknowledge financial support from the Netherlands Organization for Scientific Research (NWO) and the ECMWF for computational time on their supercomputers.



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

## Appendix A: Scale analysis

Using scale analysis on the heat equation, we estimate the depth heat diffusion reaches within a model time step:

$$\frac{\partial T}{\partial t} = \frac{k}{\rho c_p} \frac{\partial T}{\partial z^2},$$
$$\frac{\Delta T}{\Delta t} = \frac{k}{\rho c_p} \frac{\Delta T}{\Delta z^2},$$

$$\Delta z = \sqrt{\frac{k}{\rho c_p} \Delta t}, \tag{A1}$$

with $T$ the snow/ice temperature, $\Delta z$ the depth scale, $\Delta t$ the model time step (4 minutes), $k$ the thermal conductivity, $\rho$ the snow/ice density and $c_p$ the specific heat content. Using Eq. (A1) and typical values for $k$, $\rho$ and $c_p$ (Fukusako, 1990; Sturm et al., 1997) results in $\Delta z$ to be in the order of 5 mm for both snow and ice.


## Appendix B: Figures

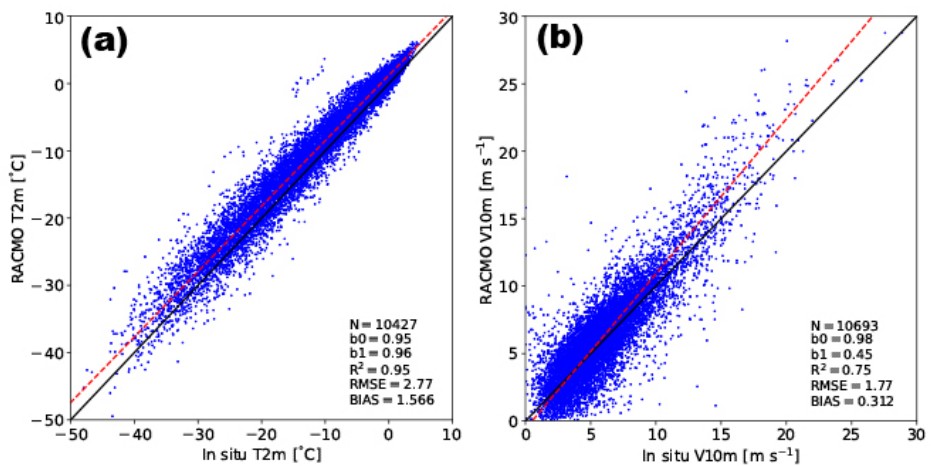

**Figure B1.** Daily-averaged **(a)** atmospheric 2-m temperature ($T_{2m}$) and **(b)** 10-m wind speed ($V_{10m}$) of Rp3 compared to in-situ observations between 2006 and 2015 for S6 and S9 (S10 is not available) of the K-transect and KAN-U and KAN-M of PROMICE. The black line is the 1-on-1 line and the red line shows the orthogonal total least squares regression of the data, with b0 the slope and b1 the intercept. Number of records (N), correlation coefficient ($R^2$), root-mean-square error (RMSE, in $^\circ$C and m s$^{-1}$, respectively) and bias are displayed.



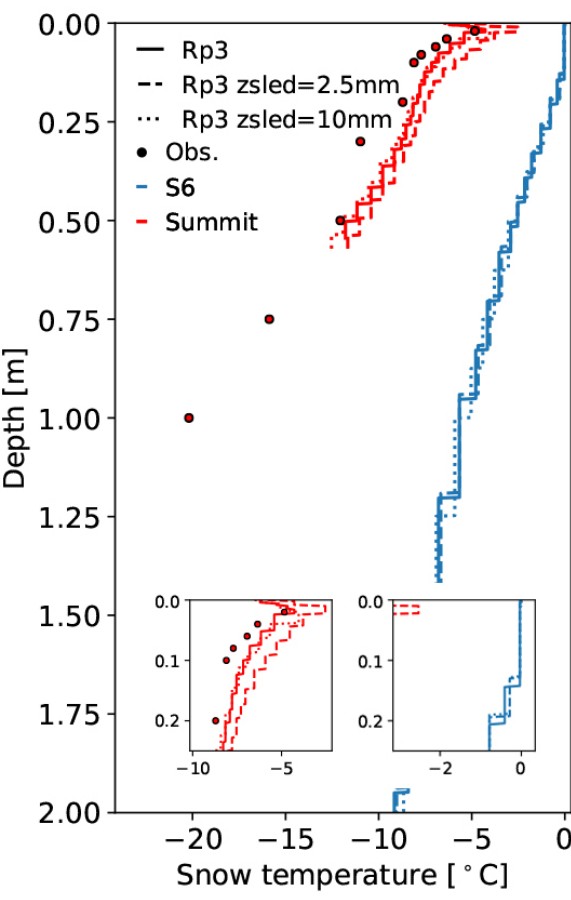

**Figure B2.** Subsurface temperature profile for the 20 upper snow layers of Rp3, Rp3 with $z_{sled} = 2.5$ mm, Rp3 with $z_{sled} = 10$ mm and observations (Obs.) for S6 and Summit for a summer day (10 July 2007) at 15:00 UTC. The insets show the results of the upper layers in more detail.