# Peer review of "Impact of updated radiative transfer scheme in RACMO2.3p3 on the surface mass and energy budget of the Greenland ice sheet"

_The Cryosphere, 2020_

## Short Comment (SC1) · 23 Oct 2020

This study is a valuable contribution to improving modelled surface energy balance of ice sheets. RACMO is a widely used climate model on both the Greenland and Antarctic ice sheets, and developing a model scheme to account for subsurface energy absorption is important. I find the manuscript well-written and the study is comprehensive in many regards. It provides much insight into radiative energy absorption, which has so far been neglected in regional climate models and firn models. The main objective of this short comment is to support the publication of the study, to provide a few suggestions for improvement and to ask questions about the findings. My suggestions

are only minor, and I know that a future comprehensive evaluation by reviewers will be more substantial and will address several issues that I do not mention.

For readers who are not familiar with surface energy balance and radiative energy transfer, it can be complicated to grasp all components at play. Part of the radiative energy is directly re-emitted, part of it contributes to the surface energy absorption, part of it contributes to the internal energy absorption and this partitioning depends on both wavelength and material properties. In my opinion, Figure 9 helps to how these aspects intertwine. But it comes rather late in the manuscript and it may be difficult for the reader to have a clear understanding of the radiative transfer when evaluating the results. I think it would be helpful to have a graph similar to Figure 9 alongside Figure 1, maybe for theoretical snow/ice layers and radiation conditions. Alternatively, a comprehensive table could also help. Three columns could be used ("Reflected", "Surface absorption", "Internal absorption"), with two rows ("Fresh snow", "Ice"). Each row can subsequently be split into three subrows ("UV", "Visible", "Near-IR"). The columns for each subrow can then be filled with theoretical % values or qualitative estimations ("Very low", "Low", "Medium", "High", etc.).

The impact of internal energy absorption on melt in the interior should be better explained. I find the explanations provided a little contradictory: Section 3.1: " In the interior, significant differences are observed only in south Greenland, with a considerable increase of snow melt (Fig. 3a). As radiation penetration now allows internal heating, it consequently raises subsurface temperatures and increases melt." Section 6.2: " In a broad elevation band around the center, less snow melt and refreezing is observed, illustrating the importance of subsurface energy absorption, which prevents the surface from reaching the melting point." Concerning the statement in Section 3.1, Figure 12b shows that melt differences in south Greenland are essentially albedo-driven and that the contribution of internal heating is minor (as explained at the end of Section 6.2). I understand that there might be a delicate balance between the skin temperature not reaching the melting point and extra subsurface melt. The distinction between dry and

melt conditions is important for many reasons, and I hope that the authors can better quantify the isolated impact of internal energy absorption on melting in interior Greenland (which would also be crucial for large areas in Antarctica). Figure 12b shows that internal heating increases melt in the interior. This contradicts the statement of Section 6.2. However, in some localised interior areas, internal heating decreases melt. Why do such differences occur in similar climatic conditions?

The authors have demonstrated how subsurface melt can liberate pore space in ice layers and subsequently increase refreezing capacities. This is an important point with respect to meltwater infiltration and SMB modelling. What can the authors say about the physics of that process? Is it physically plausible that subsurface melt makes ice layers more porous? Or is this simply a model artefact? I understand that the density of an ice layer affected by subsurface melt decreases in Rp3. However, could it be that it is the ice layer thickness that decreases in reality (its uppermost part melts away)? In this case, no additional porosity would be generated and runoff may be enhanced. This is a genuine question.

Finally, here are a few technical comments. I think there might be a sign error in Figure 1. Since F(z) is decreasing with depth, should dF(z)/dz not be negative at all depth? Values on the y-axis are shown as positive however. Can the authors consider changing Figure 12 from (Rp3 WIE - RP3) to (RP3 - RP3 WIE)? This would make a comparison between Figures 3 and 12 more straightforward, so that the reader has a clearer picture of changes related to the modifications between (i) Rp2 and Rp3 and (ii) Rp3 WIE and Rp3. I think that "temporal variability" should be replaced by "spatial variability" on line 234. I believe that there is a typo on line 297. If I understand correctly, it is a lower fraction that contributes to the SEB at S6 compared to Summit and not a "larger fraction". It would be interesting to plot more than the 20 upper layers of Rp3 in Figure 10b to allow a better comparison of the model with the observations. There are some typos/repetitions in Section 6.3.

I wish the authors good luck with the publication process.

---

## Referee Comment (RC1) · Anonymous Referee #1 · 10 Nov 2020

**A review of « Impact of updated radiative transfer scheme in RACMO2.3p3 on the surface mass and energy budget of the Greenland ice sheet » by van Dalum, et al., submitted to *The Cryosphere*.**

**Overview**

This manuscript presents the updated radiative transfer scheme of RACMO2.3p3, its impact on surface mass balance (SMB), and surface energy budget (SEB). The updated scheme enables the representation of subsurface warming causing by radiation penetration.
Considering these new developments, the results of SMB, SEB, and their respective components are compared with various in-situ observations (automatic weather station of K-transect, PROMICE and temperature profile at Summit) and the former version of the model (RACMO2.3p2).
The SMB representation is improved, compared to the previous version of the model, in the percolation zone and more generally around the ice margin. Comparison of subsurface temperatures in different snow layers at Summit is good. This updated radiative transfer scheme enables therefore to improve the representation of the subsurface melt extension.

The authors correctly evaluate an original radiative transfer scheme in RACMOp. The manuscript is very interesting and I'm curious to read what this brings to the RACMOp projections. However, I raised many points that should be considered to improve the manuscript. Since they mainly concern the presentation and form, I recommend a minor revision as it doesn't request further developments or experiments.

**General comments**

This study first describe and evalue the new and original representation of a physical process trough an updated radiative scheme in RACMOp. Without discussing anything on the scientific background of this manuscript, in its current state, the way of expressing ideas sometimes does not enable an immediate understanding. Indeed, some passages require several readings for a complete understanding. For instance, in section 4, the links between figures and explanations or justifications of processes present are not very clear and could be improved and better explained. A series of comments were made in an attempt to fill this lack of immediate understanding throughout the manuscript (see specific comments).

Moreover the structure of the manuscript is rather surprising. The updated version of the radiative scheme is partially evaluated in van Dalum et. al. (2020, published in TC), but the part concerning radiation penetration was not evaluated there. After reading the title and the abstract, I mainly expected in the present to see an evaluation of RACMO with this new radiative scheme, and in a second step a comparison of the two budgets to assess their respective influence. However, the manuscript is constructed the opposite way which might be not very intuitive in my opinion. I would therefore suggest to first bring the sensitivity experiment which represents an evaluation of the radiation penetration part of the radiative scheme alone and describe the internal energy absorption (what its presented in the title), then to show its impact on the subsurface temperature profile, on the SEB and finally on the SMB and its components compared to previous RACMO versions.
Indeed, discussing and describing the internal energy abortion regionally (as for example in paragraph 5.1) firstly would help to provide a basis for the subject discussed throughout the results, and would undoubtedly lead to a better understanding on first reading.
Several times the authors insist on the evaluation of the new albedo and radiative transfer scheme, but the part concerning albedo is mainly evaluated in a previous paper (van Dalum et al., 2020, published in TC). Although closely related, it is obvious that links have be made with this first paper. However, one would not expect to see an evaluation of the new albedo scheme in this

manuscript (despite physical connections between the albedo and radiative scheme). Yet the distinction between the two is weak in the abstract, in the introduction and even in the methodology. Furthermore, comparing RACMO2.3p2 with RACMO2.3p3 (main parts of the manuscript) amounts to compare both improvement in the radiative scheme and albedo preventing the reader to assess "the impact of (the) updated radiative transfer scheme in RACMO2.3p3 on the surface mass and energy budged of the Greenland ice sheet".

In the introduction, the author could relate the new radiation scheme more to other works and references: Has this already been done with another climate model? What is the relevance of using this new scheme and not another one? (...?) In general, the manuscript could benefit from adding more broad-scale context and impacts. While exhaustive evaluation of climate models are required and welcome to know the biases of the models (to be put in perspective with the projected changes), the broader scientific interest/question could be more detailed to move away from the papers that can be found in Geoscientific Model Development (ie, only evaluation without any/very few scientific discussion).

In order to bring more scientific credibility to the manuscript, I highly advise authors to reformulate each comparison described in the text, and to evaluate its significance using a simple statistical test (student test for example) or at least a comparison of statistical variability (RMSE) to the observed biases. Each time the authors qualify an increase/decrease/... terms with poor scientific value ("considerable", "important",...) are used that do not reflect scientific rigour. This comment is valid for all the variables studied, whether they concern the SMB, the SEB, the temperature. I would suggest the author to define at least a threshold for stating about "important" changes.

*Specific comments*

- P. 1, L. 1: The abstract requires an introduction sentence to situate this work in its broader scientific context (also true for its conclusion). The authors should also try to introduce in the abstract what the study brings in a wider scientific context than the internal and technical improvement of the model used.

- P. 1, L. 3: « […] as subsurface heating by radiation penetration now occurs ». I suggest to reformulate for instance like this: « […] as the representation of radiation penetration enables to simulate subsurface heating. »

- P. 1, L. 18: « Snow and ice melt typically dominate the SMB [...] », in your SMB equation (2) on P. 4, melt is not a component of the SMB and can't then dominate SMB? Since it rather runoff that dominates the SMB over these areas, this sentence should be rephrased.

- P. 2, L. 34: Concerning melt, don't the downscaling techniques, commented above (statistical downscalling), enable us to avoid problems linked to poorly represented topography with a coarse resolution? Doesn't this lack of precise representation of the topography affect the SMB precipitation component more than the runoff component (and associated melting)?

- P. 2, L. 36-42: Throughout this paragraph, the theoretical description of radiation penetration and its influences is well written. However, the link with modeling is poorly introduced. For instance, I suggest  to move this sentence: « Parametrizations of radiative … (Fettweis et al., 2017). » to place it at the end of the paragraph, and rephrase it by adding information about how these processes are now represented (or not) in RCMs.

- P. 4, eq (2): Authors detailed runoff components, but not erosion (ER) ones. Following the SMB equation over Greenland in Lenaerts et al. (2012, 2014), drifting snow (DS) erosion is also associated with DS sublimation. Could you clarify the different components in your equation (2)?

- P. 4, L. 99-100: Radiative scheme is called each hour. Have some sensitivity experiments been carried out on the call frequency of the radiative scheme? Does this have an influence on SEB and subsurface heating results?

- P. 4, L. 109: Could you specify the order of magnitude or the average height of the snow/ice layers considered for internal energy absorption and SEB?

- P. 5, L. 137: Has an evaluation of the model performance forced by the ERA5 reanalysis already been carried out? Could this influence/improve the results of the new radiation scheme evaluation?

- P. 8, L. 167: What elevation difference threshold did you choose to not use an AWS?

- P. 8, L. 170: For a homogeneity of the method, I suggest to chose a single RACMO grid point, the closest one, for comparison with SMB and SEB observations, as for subsurface comparisons.

- P. 8, L. 176: Please specify that you are talking about annual SMB, in the main text and in captions of your figures even if the units give some clues on this (also for SMB components).

- P. 8, L. 178: Which statistical parameter corresponds to 20 mm w.e. $yr^{-1}$? It would make more sense to use a local statistical comparison given the large variation in intensity of the components of SMB and SMB (20 mm w.e. $yr^{-1}$ seems to be high over the centre of the ice sheet but particularly low over margins where SMB has more viariabily). As suggested in general comments, a t-test or a comparison to the interannual variability of each grid point (RMSE) could be performed. This could be plotted on the spatial representations of the results as hatched if the results are significant or not. This will also strengthen the conclusions of the manuscript.

- P. 8, L. 181: Integrated SMB is usually given in Gt $yr^{-1}$.

- P. 8, L.186: « a strong SMB increase » Please specify that you are referring to Figure 2b.

- P. 8, L188: « The outer rim of the ice sheet, except in the southeast, is characterized by a strong SMB increase. In Rp2 at 11 km, the bare ice albedo of the majority of the outermost glaciated points is 0.30 due to contamination with tundra albedo, causing too much melt and runoff. This artifact is mitigated for higher resolutions and is solved in Rp3 (Van Dalum et al., 2020), lowering the snow melt and runoff and increasing the SMB. »
When reading this paragraphs, I understand that this bias is mostly caused by the spatial resolution (mix between tundra and ice albedo). The lower also leads to a surestimation of associated melt and runoff. For me, this is not corrected by the improvements in Rp3. If the runoff biases really comes from the resolution, and since both Rp2 and Rp3 simulations are performed at the same resolution, it would be better to state that the biases is compensated by the improvements of the radiative scheme and not mitagtaed (since you have'nt corrected the resolution that leads to the contamination)..

- P. 9, L. 194: Why subsurface melting of ice creates pore space? Could you more explain this affirmation and not just refer to van Dalum et al (2020). It deserves to be clearly explained, especially since it is repeated in the conclusion. This result seems counterintuitive since melt leads to an increase in density and thus to a reduction in available space.

- P. 10-11, subsection 3.2 and Figure 4: If you consider melt events (and not per year), why do you chose a threshold per year (250 mm yr$^{-1}$ in the caption of the Figure 4)?

- P. 10, L. 210: Results averaged over ice sheet should be given in Gt yr$^{-1}$, same comment than in L. 181.

- P. 13, L. 240: Note that R² is the abbreviation for determination coefficient of a regression line (that expresses the part of a variable that can be expressed by another) while the coefficient correlation (that associates variations of two variables) is noted by R. Annotation (R² to R) or the use of a non-adapted variable (correlation to determination) should also be corrected in the Table 1, Figure 5, 6 and B1 captions depending on what you meant.

- P.13, L. 241-243: « There is a tendency for Rp3 to underestimate LWd during cold and dry, cloud-free winter (small SWd) conditions, resulting in underestimated LWu for the same cold days. » Please could the authors clarify these explanations and what they are basing this reasoning on?

- P. 14, Figure 7: Do you have an idea of why Rp3 improves SHF of Rp2 during Jul-Aug-Sep at S6? (while Rp3 has poorer results for this variable in the rest of the comparison)

- P. 15, L. 259-261: « The introduction of a new radiation penetration and snow albedo scheme reduces the differences with observations for Rp3 compared to Rp2. This is especially noticeable at the onset of the accumulation season [...] »: Is this amelioration statically significant? At what point is improvement considered? Do you have hypotheses to explain these discrepancy (especially in summer for SWn)?
More generally, comparison between Rp2 and Rp3 (mainly Table 1 and Figure 7) suggest a relatively limited amelioration (and even deterioration of the results) poorly discussed in the text.

- P. 16, L. 296-297: « For bands 7 to 12, almost all incoming radiation is absorbed at S6 due to the large grain radius and density. For these bands a larger fraction of energy contributes to the SEB at S6 compared to Summit. » : Figure 9 for Bands 7 to 12 at S6 suggests the opposite, the authors would probably write « a lower fraction » instead of « a larger fraction »?

- P.17, L. 308 and 309: Please specify what do you mean by « typical winter » (cloud, precipitaitons, …?) and « extrodinary warm summer day ».

- P. 21, L. 377: Please justify negligible.

- p. 22: Several times in the results sections, Rp3 reveals a weakness in the representation of turbulent fluxes (LHF and SHF). This deserves a point of improvement in the model in your conclusion.

_Tables and Figures_

- Figure 2: « Subsurface temperature profiles are available for Summit » Please specify that is the green dot.

- Table 1: Please align the numbers to the right for better visibility ; Add unit for bias in the legend.

- Figure 5: Please add bias and its unit in the legend.

- Figure 6: Please add unit for bias in the legend.

- Figure 7: For more visibility, could you enlarge or thicken the coloured lines in the legend (especially for LWn, SHF and LHF)?

- Figure 7: At certain places in the figure, the values of the three time series are similar so that it's complicated to distinguish them. This is not a problem when they are all three grouped together, but when two are grouped together and not the third one, it is difficult to distinguish which one is hidden behind the other. This can lead to a misunderstanding of the figure.

- Figure 10c: Please justify why this specific grid point is choosen (SE GRL, star on Figure 2).

*Additional references*

P. 2, L. 32: « The SMB components can be statistically downscaled to an even higher resolution of up to 1 km (Noël et al., 2019) ». The authors can add these two references about dwonscalling :

Franco, B., Fettweis, X., Lang, C., and Erpicum, M.: Impact of spatial resolution on the modelling of the Greenland ice sheet surface mass balance between 1990–2010, using the regional climate model MAR, The Cryosphere, 6, 695–711, doi:10.5194/tc6-695-2012, 2012

Fettweis, X., Hofer, S., Krebs-Kanzow, U., Amory, C., Aoki, T., Berends, C. J., Born, A., Box, J. E., Delhasse, A., Fujita, K., Gierz, P., Goelzer, H., Hanna, E., Hashimoto, A., Huybrechts, P., Kapsch, M.-L., King, M. D., Kittel, C., Lang, C., Langen, P. L., Lenaerts, J. T. M., Liston, G. E., Lohmann, G., Mernild, S. H., Mikolajewicz, U., Modali, K., Mottram, R. H., Niwano, M., Noël, B., Ryan, J. C., Smith, A., Streffing, J., Tedesco, M., van de Berg, W. J., van den Broeke, M., van de Wal, R. S. W., van Kampenhout, L., Wilton, D., Wouters, B., Ziemen, F., and Zolles, T.: GrSMBMIP: Intercomparison of the modelled 1980–2012 surface mass balance over the Greenland Ice sheet, The Cryosphere Discuss., https://doi.org/10.5194/tc-2019-321, accepted, 2020.

- P. 2, L. 33: I suggest the authors to refer to Fettweis et al. (2020) for a more detailed RCM evaluation from different models.

---

## Referee Comment (RC2) · Anonymous Referee #2 · 18 Nov 2020

The authors evaluated the effect of a new snow/ice albedo and radiative transfer treatment in RACMO2 model over the Greenland ice sheet by comparing with several different in-situ measurements. They found that the modeled surface mass balance generally agrees with observations and the new snow/ice albedo treatment has a nontrivial impact on the surface mass balance. The paper is generally well-structured and suitable for this journal. Before it can be considered for potential publication, I have a few comments and suggestions for the authors to consider. Particularly, there are several places that still need more clarifications.

Specific comments: 1. Title: I suggest including "snow and ice" before "radiative transfer scheme" to avoid potential confusion.

2. Abstract (Line 6): "The surface mass balance is in good agreement with observations". Please be more quantitative, e.g., bias within a few percent?

3. Lines 68-71: Does the update of multilayer firn module only includes an increase in the number of layers? How about related physical processes? Any updates on the physics? More clarifications would be good.

4. Lines 73-82: Although the authors mentioned that their recent paper (Van Dalum et al. 2020) provided a detailed description of the new snow albedo module, it would be good and informative to include some key elements of this new snow/ice albedo module. For example, as partially mentioned in the introduction, snow grain properties (size, shape), snow impurities, and snow layer structures are critical to snow albedo calculation. How does the new albedo module deal with these elements? Also, more descriptions of the bare ice albedo parameterization are needed. How does the module handle ice underlying snow layers, e.g. assuming a fixed ice albedo or explicitly resolve radiative transfer within ice layers below snow? Besides, what snow and ice processes have been included in this coupled RACMO2 model, e.g., sublimation, redistribution, snow grain growth, refreezing, etc?

5. Lines 89-97: How does the model deal with the Gs (subsurface conductive heat flux) at the ice surface below snow layers? Or did the authors only consider Gs at the soil/ground surface below both snow and ice layers? Does the runoff only include water coming out of snowpack above the underlying ice? Is there lateral water flow between neighboring grids in the model? A more straightforward question would be whether the model deals with the permanent ice glaciers under snow layers? If so, how?

6. Section 2.3: The authors calculated internal energy absorption at non-FR time steps by using a simplified Beer's law (i.e., exponential decay) equation. Would this cause any discontinuity of the absorption profiles between non-FR and FR time steps since the TARTES is used for FR time steps?

7. Section 2.5: What are the observational uncertainties of these in-situ measurements?

8. Section 3.2: This part is very interesting, however, I am not quite convinced that the rather small snow melt in the central GrIS could have such a large impact on the cloud cover in the region. Any explanations on the mechanisms? Did the authors see an increase in surface heat and water vapor sources due to snow melting and albedo increase, which could enhance the cloud formation locally? Also, snow melting could also decrease snow albedo, so I am not sure if the change in spectral distribution of irradiance is large enough to cause the snow albedo increase as mentioned by the authors. More explanations and clarifications are needed.

9. The authors conducted model evaluation by comparing with several in-situ point measurements. I suggest including some regional scale evaluation of the model, particularly surface albedo, by comparing with satellite products (e.g., MODIS). Large uncertainty could exist in the comparison of 11-km model grid data with point observations.

---

## Editor Comment (EC1) · Masashi Niwano (Editor) · 19 Nov 2020

Dear authors, cc: referees,

Now, you have received two referee reports and one short comment. I would like to thank the reviewers and Vincent Verjans for providing constructive comments/suggestions. I believe they will increase the quality of the manuscript certainly, so I hope that you revise the manuscript considering these comments/suggestions thoroughly.

I also list my specific remarks as follows:

[Figure]

Title: I agree with the comment by Referee #2 on the current title. When I first read the title, I imagined you would discuss a radiative transfer scheme in atmosphere; however, it turned out that the main purpose of this study is to discuss impacts of a radiative transfer scheme for snow and ice. Can you specify in the title that you focus on a radiative transfer within snow and ice in this present study?

Sect. 6.3: I agree with Vincent Verjans that the discussion in Sect. 6.3 is a bit difficult to follow due mainly to typos. Please check and reformulate the subsection carefully.

Throughout the manuscript, you explicitly indicate "snow melt". But do you really want to distinguish snow melt and ice melt in this study? I think indicating just "melt" throughout the manuscript may be enough.

L. 23: The intention of "darkening of the snow pack" is unclear. Please clarify.

L. 36: "ground" -> "subsurface conductive"

L. 79: Please define Bands 13 and 14 here.

L. 116: "(4 minutes)" should be indicated earlier in this section, e.g., L. 100?

L. 107: The direction of the flux $F(z)$ should be introduced.

L. 118: More detailed explanation of "maximum skin layer equilibration depth (SLED)" is needed.

L. 143: For "grain radius", you should specify whether it is geometrical or optical. This is also the case for other parts that mention snow grain size. For example, this point is particularly important when readers see Figure 8.

L. 147: For Rp3 WIE, you should mention whether the internal energy absorption is discarded or added to the surface; I realized that this point is explained at L. 368. But L. 147 may be the better place to explain that.

L. 177: "The SMB changes are typically significant with respect to the inter-annual

variability if they exceed 20 mm w.e. yr-1.": From which figure can we see this?

L. 254; "net absorbed longwave radiation" -> "net longwave radiation"

Figure 7: It is difficult to distinguish Rp2 and Rp3 in this figure. Can you update the figure? A similar comment is provided by Referee #1.

L. 258: "net absorbed shortwave radiation" -> "net shortwave radiation"

L. 260: Please specify "the onset of the accumulation season".

Figure8 caption: "(a), (b) Grain radius and (c), (d) absorption of solar radiation" -> "(a), (b) Grain radius and (c), (d) absorption of solar radiation simulated by Rp3"

L. 288: "visible light" -> "UV and visible light"

L. 294: "drops" -> "reductions"

L. 308: "temperature profile" -> "temperature profiles"

Figure 10 caption: "temperature profile" -> "temperature profiles" * There are two places.

L. 329: "temperature profile" -> "temperature profiles"

L. 349: Please consider adding a reference for this argument.

L. 367: "the SMB, snow melt and refreezing difference" -> "differences in the SMB, snow melt and refreezing"; Please also check whether "snow melt" is an appropriate word or not as mentioned above.

L. 369: "melt" -> "meltwater"

L. 372 ∼ 373: "More importantly, however, is that internal radiative heating reduces refreezing in the ablation zone (Fig. 12c), especially in the southwest, but also in the northwest and northeast (area C, D and B of Fig. 2c, respectively), so that runoff increases, reducing the SMB.": This sentence seems strange to me. Please check the

sentence as well as Fig. 12 carefully. It seems to me that the SMB in the ablation area is simulated to be higher for the Rp3 than Rp3 WIE.

L. 372: "area C, D and B" -> "areas A and B"

L. 373: "melt" -> "meltwater"

L. 378 "albedo decrease" caused mainly by what?

L. 403: "resolution" -> "horizontal resolution"

Appendix A: The first diffusion equation contains a typo.

---

## Author Comment (AC1) · 28 Jan 2021

**Response on the referee comments on the manuscript:**

Impact of updated radiative transfer scheme in RACMO2.3p3 on the surface mass and energy budget of the Greenland ice sheet by C.T. van Dalum et al.

We would like to thank the reviewers for their constructive comments that have improved the accuracy of the evaluation and the clarity of the paper. In black the comment, in orange the response, in blue the changes that we would implement in the manuscript.

We have restructured most of the manuscript, almost all lines and figures are now located on a different page. Here, we always refer to the pages, sections, figures and lines as can be found in the manuscript that you reviewed, not to the new version.

We discuss the comments in the following order: reviewer #1, the editor comments, the comments by Vincent Verjans and finally the comments of reviewer #2. Lastly, the revised Figures of this manuscript are provided. There is no specific reasoning behind this order, as the order is irrelevant as long as all comments are addressed.

**Review #1**

**General comments**

This study first describes and evaluates the new and original representation of a physical process through an updated radiative scheme in RACMOp. Without discussing anything on the scientific background of this manuscript, in its current state, the way of expressing ideas sometimes does not enable an immediate understanding. Indeed, some passages require several readings for a complete understanding. For instance, in section 4, the links between figures and explanations or justifications of processes present are not very clear and could be improved and better explained. A series of comments were made in an attempt to fill this lack of immediate understanding throughout the manuscript (see specific comments).

See specific comments for the changes, and also the last comment of Vincent Verjans and comment 4 of reviewer #2.

Moreover the structure of the manuscript is rather surprising. The updated version of the radiative scheme is partially evaluated in van Dalum et. al. (2020, published in TC), but the part concerning radiation penetration was not evaluated there. After reading the title and the abstract, I mainly expected in the present to see an evaluation of RACMO with this new radiative scheme, and in a second step a comparison of the two budgets to assess their respective influence. However, the manuscript is constructed the opposite way which might be not very intuitive in my opinion. I would therefore suggest to first bring the sensitivity experiment which represents an evaluation of the radiation penetration part of the radiative scheme alone and describe the internal energy absorption (what its presented in the title), then to show its impact on the subsurface temperature profile, on the SEB and finally on the SMB and its components compared to previous RACMO versions. Indeed, discussing and describing the internal energy abortion regionally (as for example in paragraph 5.1) firstly would help to provide a basis for the subject discussed throughout the results, and would undoubtedly lead to a better understanding on first reading.

As requested, we changed the structure of the manuscript. After the method section, we now first discuss the subsurface energy absorption (so paragraph 5.1 and 5.2). This is followed by the temperature analysis of paragraph 6.1 and 6.2. Then, we discuss the surface energy budget, followed by the surface mass balance section. The SMB section is ended with paragraph 6.3, so the comparison of the SMB (components) with Rp3 WIE. Of course, the figure order is changed accordingly, and the locations of Fig. 2b are now moved to Fig. 11c, which is now the first Figure with a map. To summarize, the section order is now as follows.

Sect. 3: Subsurface energy absorption.
3.1 K-transect
3.2 Distribution of energy
Sect. 4: Temperature experiments
4.1 Subsurface temperature
4.2 Snow temperature at 10 m depth
5. Surface energy budget
6. Surface mass balance
6.1 Comparison with RACMO2.3p2
6.2 Cloud cover and melt
6.3 SMB observations
6.4 SMB without internal energy absorption
7. Conclusions

In addition, several changes are made in the text to better follow the new structure.

**Page 2:**

In this paper we assess the impact of the improved radiative transfer scheme in snow and ice on the modeled SMB and SEB for the GrIS in Rp3. Section 2 discusses the model and initialization, expands on the concepts of SMB, SEB and internal energy absorption, and discusses the in-situ data sets in more detail. In Section 3, internal energy absorption due to radiation penetration is assessed. In Section 4, sensitivity experiments further highlight the impact of internal heating on the subsurface temperature. Results are compared with observations and compared to the previous RACMO2 version, 2.3p2 (Rp2) (Noel et al., 2018) and to a Rp3 model version without internal energy absorption (Rp3 WIE). In Section 5, results are analyzed by evaluating the SEB to observations and by comparing modeled SEB of Rp3 with Rp2. Similar to Section 5, Section 6 focuses on the SMB and its components. We also compare the SMB of Rp3 with Rp3. WIE. Finally, Section 7 summarizes the results and conclusions are drawn.

**Page 7**

...accumulation sites (shown in Fig. 11c)...

**Page 11**

A large fraction of UV and visible radiation is absorbed internally (Fig. 9)...

**Page 15**

...the K-transect for 2012 in Fig. 8 (locations illustrated in Fig. 11c).

**Page 17**

Renamed the section to: Temperature experiments

**Page 17**

Compared to observations, the 2-m air temperature difference is only small (Table 1 and Fig. B1). The impact of internal energy absorption on the subsurface temperature, however, is larger for various locations in winter and summer, and is discussed in this section. We also compare the results with Rp3 WIE. Furthermore, we discuss the 10-m snow temperature (T10m). We also investigate the impact of zsled on the subsurface temperature, which is shown in Appendix B, Fig. B2.

**Page 18 & 19**

(SE GRL, star in Fig. 11c)

**Page 21**

Finally, we discuss the impact of radiation penetration...

**Page 22**

Moreover, subsurface temperature profiles show a more gradual temperature decrease with depth in Rp3 during summer in the interior, improving the agreement with observed snow temperatures at Summit; and melt reaches deeper layers both in the accumulation and ablation zone.

Several times the authors insist on the evaluation of the new albedo and radiative transfer scheme, but the part concerning albedo is mainly evaluated in a previous paper (van Dalum et al., 2020, published in TC). Although closely related, it is obvious that links have be made with this first paper. However, one would not expect to see an evaluation of the new albedo scheme in this manuscript (despite physical connections between the albedo and radiative scheme). Yet the distinction between the two is weak in the abstract, in the introduction and even in the methodology. Furthermore, comparing RACMO2.3p2 with RACMO2.3p3 (main parts of the manuscript) amounts to compare both improvement in the radiative scheme and albedo preventing the reader to assess "the impact of (the) updated radiative transfer scheme in RACMO2.3p3 on the surface mass and energy budged of the Greenland ice sheet".

You are right that sometimes we did not present it clearly that we do not want to evaluate the albedo scheme here. However, we do mention on several occasions that we do not only want to evaluate the radiative scheme, but also want to look into the impact that all changes have (including the albedo) on the surface mass balance and energy budget. So it is not always possible to clearly distinct between albedo and radiative transfer, as some changes in the SMB or SEB components are mostly caused by albedo changes. We improve the manuscript on a few places. For the changes of Sect 2.1: see comment 4. of review #2.

**Page 1**

In this study, we evaluate a new englacial radiative transfer scheme and assess the surface mass and energy budget for the Greenland ice sheet in the latest version...

**Page 2**

In this paper we assess the impact of the improved radiative transfer scheme in snow and ice on the modeled SMB and SEB for the GrIS in Rp3. Section 2 discusses...

**Page 22**

Regional climate models often do not explicitly account for radiative transfer in snow and ice. Therefore, we evaluated the latest version of the regional climate model RACMO2, which has an updated radiative transfer scheme that allows internal heating. In this new version Rp3, we also updated the multilayer firn module and snow and ice albedo (Van Dalum et al., 2020), and developed a method to distribute total absorbed shortwave radiation between the SEB and to internal absorption. In this study, we assessed the modeled surface mass balance (SMB), surface energy budget (SEB) and firn and snow temperature of the GrIS.

In the introduction, the author could relate the new radiation scheme more to other works and references: Has this already been done with another climate model? What is the relevance of using this new scheme and not another one? (...?) In general, the manuscript could benefit from adding more broad-scale context and impacts. While exhaustive evaluation of climate models are required and welcome to know the biases of the models (to be put in perspective with the projected changes), the broader scientific interest/question could be more detailed to move away from the papers that can be found in Geoscientific Model Development (ie, only evaluation without any/very few scientific discussion).

We have added a new paragraph in the introduction to discuss how radiative transfer is handled in other RCMs. Furthermore, we added a lot of explanation in the method section about the updates in Rp3 and why it is relevant (See comment 4 of review #2).

**Page 2:**

...subsurface energy absorption can lead to subsurface melting if the surface is close to or at the melting point.

Parameterizations of radiative fluxes can still be improved in RCMs. The Modèle Atmosphérique Régional (MAR), for example, uses the Surface Vegetation Atmosphere Transfer scheme SISVAT (Gallee et al., 1991; De Ridder and Gallee, 1998; Fettweis (2007); Fettweis et al., (2017). SISVAT does not contain a proper radiative transfer scheme in snow or ice; the albedo is therefore parameterized

by using various tuning parameters. Internal heating by shortwave radiation is also not calculated and is therefore not included in the heat balance of a layer. Similarly, the RCM HIRHAM5 (Lucas-Picher et al., 2012), which uses the physics of the ECHAM5 model (Roeckner et al., 2003), does not consider radiative transfer in snow and ice, and the snow albedo is calculated simply as a linear function with the surface temperature. The RCM that we use in this study, the polar (p) version of the Regional Atmospheric Climate Model (RACMO2), also does not include radiation penetration.

**RACMO2 has been used extensively to model large scale, ice-sheet wide developments...**

In order to bring more scientific credibility to the manuscript, I highly advise authors to reformulate each comparison described in the text, and to evaluate its significance using a simple statistical test (student test for example) or at least a comparison of statistical variability (RMSE) to the observed biases. Each time the authors qualify an increase/decrease/... terms with poor scientific value ("considerable", "important",...) are used that do not reflect scientific rigour. This comment is valid for all the variables studied, whether they concern the SMB, the SEB, the temperature. I would suggest the author to define at least a threshold for stating about "important" changes. We did not realize that we used those words so often without a scientific explanation. So we have looked at every occasion where we use words like 'considerably' or 'important', and have removed some that are unnecessary. For all other occasions, we have put quite some effort into investigating the significance by using statistical bootstrapping, or justify it by other means. We have also added hatched areas in some maps, i.e., figures 2, 3 and 12 (See end of this document for the Figures). Furthermore, we calculated the significance of the bias compared to observations of the SEB components and SMB. Except for LWd, all SEB components show a statistical significant bias. However, the measurement uncertainty is often higher than the BIAS. The bias with observations for SMB are statistically insignificant. The few occurrences in the introduction are not altered, as they refer to other work.

**Page 1**

Internal heating leads to higher snow temperatures..

**Page 5**

...more energy is absorbed within the SLED...

**Page 5**

To determine the statistical significance of the bias between model versions or the bias with observations, we use statistical bootstrapping with a significance of two standard deviations.

**Page 7**

...is used and 20 mm w.e. yr-1 elsewhere. The hatched areas show statistical significance. (c) The relative difference...

**Page 8**

...as the size and depth of snow layers in RACMO2 can alter considerably between two...

**Page 8**

Fig. 2b also shows the statistical significance. Almost all SMB changes are significant with respect to the inter-annual variability, only some of the smaller changes in the interior are insignificant. Integrated over the ice sheet, the SMB increases by 17.0 mm w.e. yr-1 (29.1 Gt yr-1), or 9.7%. The annual difference between Rp3 and Rp2 and significance for melt, refreezing and runoff are shown..

**Page 8**

...inducing internal heating. As is discussed in Sect. 5, internal heating raises the subsurface temperature and increases melt. Because all...

**Page 8**

... is characterized by a strong and significant SMB increase...

**Page 9**

...runoff. The hatched areas in (a)-(c) show statistical significance.

**Page 22**

...runoff. The hatched areas in (a) show statistical significance.

**Page 9**

As a result, the increase in melt and refreezing balance out, leading to an insignificant runoff difference for most of the southwestern ablation zone (Fig. 3), hence the SMB changes little (Fig. 2b).

**Page 10**

The larger melt extent for the star in south-east Greenland (Fig. 10c, star in Fig. 2b) is in agreement with Fig. 3d and e, where we show more melt and refreezing for this grid point. As the elevation of this location is too far above sea level, no runoff is modeled and all melt water refreezes locally. Even though the melt and consequent refreezing are still small and do not alter the SMB, they do change the snow structure.

**Page 10**

...enhanced melt leads to reduced refreezing, as the refreezing capacity...

**Page 10**

Around the K-transect, the increase of refreezing consequently lowers the modeled runoff. Significant runoff increase...

**Page 11**

For a large area of central GrIS, annual melt and refreezing have decreased by percentage... Page 12

...for Rp3 is -0.031 m w.e. yr-1 (-2.67%) and for Rp2 -0.091 m w.e. yr-1 (-7.54%), but both are statistically insignificant...

**Page 12**

Note that the spread is higher for the ablation zone, illustrating that...

**Page 13**

The influence of cloud cover is captured well, with a small and insignificant bias of LWd and a small shortwave downward radiative flux (SWd) bias. The bias of SWd, however, is still significant, as is determined by statistical bootstrapping, due to the large amount of data points. Despite its statistical significance, note that the uncertainty in the measurements (Sect. 2.5) is larger than the bias.

**Page 13**

The upward shortwave radiative flux (SWu) is also in fairly good agreement with observations. The bias, however, is still statistically significant. The bias and RMSE of these fluxes have improved in Rp3 compared to Rp2 (Table 1), confirming that the albedo has improved, which is in agreement with Van Dalum et al. (2020).

Both the sensible and latent heat flux (SHF and LHF, respectively), show a large spread and significant bias...

**Page 15**

...leading to extensive internal energy absorption (> 25 W m-2) up to 15 cm...

**Page 15**

In late June and July, firn layers that are characterized by a large optical grain radius reach the surface and induce absorption of solar radiation, although less than for bare-ice conditions. The absorption, however, is limited to the upper 5 cm. Note that the formation of a thin fresh snow layer in early July diminishes internal energy absorption.

**Page 15**

Since a fraction of the shortwave radiation is absorbed in the upper few millimeters of the snow column, not all incoming radiation contributes...

**Page 16**

...these bands is limited. For IR radiation (bands 6-12), more absorption takes place. The attenuation length...

**Page 16**

S6 has bare ice at the surface for the selected day, and more radiation is absorbed in the visible light bands due to a high concentration of soot. Most notably is the large increase of energy absorbed in band 6. This is a wide band with a considerable amount of energy available; it is also especially sensitive to albedo reductions induced by an increase of optical grain radius and density, e.g., when bare ice reaches the surface (Van Dalum et al., 2019).

**Page 18**

... are covered with fine-grained fresh snow layers. We observe only changes in the upper first centimeters, with thinner snow layers and higher temperatures in Rp3.

**Page 19**

Adding internal energy absorption increases the ability of RACMO2 to reproduce realistic subsurface temperature profiles for Summit.

**Page 19**

... is reached up to a greater depth for the accumulation point...

**Page 19**

...more melt events are expected to occur in the accumulation zone like in SE GRL. This is discussed in Sect. 3.

**Page 19**

... is small (< 0.5 degrees C) for Rp3...

**Page 19**

...however, it is considerably larger (often >2 degrees C, Fig. 11d). As all solar energy...

**Page 20**

In the percolation zone, the difference between Tskin and T10m is more substantial (more than -5 degrees C, Fig. 11b), owing to latent...

**Page 22**

...by refreezing. Close to the percolation zone close near the equilibrium line, the SMB...

**Page 22**

...the southwest, significantly more refreezing occurs due to pore space created...

**Page 22**

For the accumulation zone of the southeast, relative differences in melt and refreezing are substantial due to subsurface energy absorption and melt water percolation. This increased melt and refreezing often does not lead to significant SMB changes, but does change the snow structure.

**Page 22**

...although considerably improved in Rp3 (Van Dalum et al., 2020), could be further improved by using a dynamic Mie-scattering...

**Page 22**

... in south Greenland and close to the percolation zone...

**Specific comments**

- P. 1, L. 1: The abstract requires an introduction sentence to situate this work in its broader scientific context (also true for its conclusion). The authors should also try to introduce in the abstract what the study brings in a wider scientific context than the internal and technical improvement of the model used.

We have changed and added the following to address these issues:

**Page 1**

Radiative transfer in snow and ice is often not modeled explicitly in regional climate models. In this study, we evaluate a new englacial radiative transfer scheme and assess the surface mass and energy budget for the Greenland ice sheet..

**Page 1**

...shallow layer of subsurface melt. Hence, this study shows the consequences and necessity of radiative transfer in snow and ice for regional climate modeling of the Greenland ice sheet.

**Page 22**

Regional climate models often do not explicitly account for radiative transfer in snow and ice. Page 23

...make future climate projections. This study therefore shows the necessity of radiative transfer in snow and ice for regional climate modeling of the GrIS. In a subsequent study...

**- P. 1, L. 3: « [...] as subsurface heating by radiation penetration now occurs ». I suggest to reformulate for instance like this: « [...] as the representation of radiation penetration enables to simulate subsurface heating. »**

We changed it to the following

... as radiation penetration now enables internal heating.

- P. 1, L. 18: « Snow and ice melt typically dominate the SMB [...] », in your SMB equation (2) on P. 4, melt is not a component of the SMB and can't then dominate SMB? Since it rather runoff that dominates the SMB over these areas, this sentence should be rephrased.

**You are right**

Snow and ice melt leading to extensive runoff typically dominates the SMB around the margins of the GrIS, leading to mass loss of up to 3 m water equivalent...

- P. 2, L. 34: Concerning melt, don't the downscaling techniques, commented above (statistical downscaling), enable us to avoid problems linked to poorly represented topography with a coarse resolution? Doesn't this lack of precise representation of the topography affect the SMB precipitation component more than the runoff component (and associated melting)? Yes, many of these issues regarding resolution are solved by downscaling. Therefore, we reformulated the text.

**Page 2**

RCMs are known to perform well for most of the ice sheet, but struggle to resolve topographically inhomogeneous regions (Leeson et al. 2018). Statistically downscaling the SMB components to an even higher resolution of up to 1 km (Noel et al. 2019) solves many such issues. Still, the rough topography typically found close to the ice margin is often not modeled correctly and can lead to incorrect precipitation patterns, which illustrates the need to further improve SMB related parameterizations (Van de Berg et al., 2020).

- P. 2, L. 36-42: Throughout this paragraph, the theoretical description of radiation penetration and its influences is well written. However, the link with modeling is poorly introduced. For instance, I suggest to move this sentence: « Parametrizations of radiative ... (Fettweis et al., 2017). » to place it at the end of the paragraph, and rephrase it by adding information about how these processes are now represented (or not) in RCMs.

We have moved this sentence and have added information about other RCMs (See general comments).

- P. 4, eq (2): Authors detailed runoff components, but not erosion (ER) ones. Following the SMB equation over Greenland in Lenaerts et al. (2012, 2014), drifting snow (DS) erosion is also associated with DS sublimation. Could you clarify the different components in your equation (2)? Drifting snow sublimation is included in the SU term in RACMO2. We changed the following:

**Page 4**

...with SN snowfall, RA rain, ER drifting snow erosion, SU surface and suspended snow sublimation, and RU runoff.

**- P. 4, L. 99-100: Radiative scheme is called each hour. Have some sensitivity experiments been carried out on the call frequency of the radiative scheme? Does this have an influence on SEB and subsurface heating results?**

Unfortunately, we have not done long sensitivity experiments with another full radiation frequency, as the computational budget did not allow for it. The shorter experiments we carried out, covering a month, did not show relevant differences. Hence, we do not think that a higher frequency would result in much better results, as the radiation typically does not change drastically within an hour, and it would increase computation demands considerably. Decreasing the frequency, on the other hand, could have a large impact on the results, as, for example, a 3-hourly full radiation time interval would mean that radiation calculations would be used from noon up to 15:00 hour, which is clearly not sufficient enough.

**- P. 4, L. 109: Could you specify the order of magnitude or the average height of the snow/ice layers considered for internal energy absorption and SEB?**

The depth that some energy contributes to the SEB is described between p4, l. 114-127. For internal energy absorption, we add the following to clarify it:

**Page 4**

On average, considerable absorption of solar radiation (more than 20 W m-2) is limited to the upper 20 centimeters for clean ice, and only to several milliliters for snow.

- P. 5, L. 137: Has an evaluation of the model performance forced by the ERA5 reanalysis already been carried out? Could this influence/improve the results of the new radiation scheme evaluation? We have preliminarily results for the Antarctic ice sheet with ERA5 that show promising results and plan to run it for Greenland in the future as well. We think, however, that using ERA5 would in general only have a limited effect on the radiation scheme evaluation. Precipitation is likely the biggest factor that could be changed by ERA5, but should be relatively small. By percentage, this could result in some SMB changes in the dry interior, but is probably small in the ablation zone compared to the melt. Of course, if fresh snow is systematically located longer at the surface, it would reduce internal heating and lower the snow temperatures.

**- P. 8, L. 167: What elevation difference threshold did you choose to not use an AWS? We use an elevation differences threshold of 50 m.**

**Page 8:**

Here, we accept only grid points with an elevation differences of 50 m or less.

P. 8, L. 170: For homogeneity of the method, I suggest to chose a single RACMO grid point, the closest one, for comparison with SMB and SEB observations, as for subsurface comparisons.
You are right that it would be better for the homogeneity of the method to use the closest grid point for comparison with measurements. However, we as we use a relatively low resolution, the differences between the actual measurement station and the closest grid point can be several kilometers. In the accumulation zone, this does not matter as much, as it is a relatively homogeneous region, so with only small changes between neighboring grid points. For the ablation zone, however, this can be a problem, like at S6, as the surface conditions can vary within a few kilometers.

As we have mentioned in the text, for the subsurface grid point we use the closest grid point. This is not a problem, as we only compare it with measurements at Summit where neighboring grid points are very similar. No measurements are available for S6, however, so it isn't a problem either to not be able to interpolate between two grid points. Also, the subsurface temperature profile of S6 is only used to compare with Rp2 and Rp3 WIE, hence it is not as important to interpolate between grid points. In conclusion, we think that the least we should do to limit errors is to interpolate between grid point, so that the RACMO data is closer to the true location of the measurement stations.

- P. 8, L. 176: Please specify that you are talking about annual SMB, in the main text and in captions of your figures even if the units give some clues on this (also for SMB components). We have added 'annual' in Fig. 2, 3, 5 and 12. In addition we added it in various places in the manuscript: Page 8 ...shows the annual average 2006-2015... Page 8 The annual difference between Rp3 and Rp2... Page 10 ...the ice sheet, annual snow melt has... Page 10 The annual runoff, on the other hand... Page 10 ... while annual refreezing has increased by ... Page 11 For a large area of central GrIS, annual snow melt and refreezing have decreased Page 12 Figure 5 shows the annual SMB of Rp3 and Rp2 with respect to various SMB observations. Page 21 Figure 12 shows the annual SMB, snow melt and refreezing...

- P. 8, L. 178: Which statistical parameter corresponds to 20 mm w.e. yr-1? It would make more sense to use a local statistical comparison given the large variation in intensity of the components of SMB and SMB (20 mm w.e. yr-1 seems to be high over the centre of the ice sheet but particularly low over margins where SMB has more variably). As suggested in general comments, a t-test or a comparison to the interannual variability of each grid point (RMSE) could be performed. This could be plotted on the spatial representations of the results as hatched if the results are significant or not. This will also strengthen the conclusions of the manuscript.

We now used statistical bootstrapping to determine significance. For all changes regarding significance, see your last comment of the general comments.

P. 8, L. 181: Integrated SMB is usually given in Gt yr -1.
We have added the integrated SMB also in Gt yr-1
17.0 mm w.e. yr-1 (29.1 Gt yr-1), or 9.8%

- P. 8, L.186: « a strong SMB increase » Please specify that you are referring to Figure 2b. Done

- P. 8, L188: « The outer rim of the ice sheet, except in the southeast, is characterized by a strong SMB increase. In Rp2 at 11 km, the bare ice albedo of the majority of the outermost glaciated points is 0.30 due to contamination with tundra albedo, causing too much melt and runoff. This artifact is mitigated for higher resolutions and is solved in Rp3 (Van Dalum et al., 2020), lowering the snow melt and runoff and increasing the SMB. » When reading this paragraphs, I understand that this bias is mostly caused by the spatial resolution (mix between tundra and ice albedo). The lower also leads to a surestimation of associated melt and runoff. For me, this is not corrected by the improvements in Rp3. If the runoff biases really comes from the resolution, and since both Rp2 and Rp3 simulations are performed at the same resolution, it would be better to state that the biases is compensated by

**the improvements of the radiative scheme and not mitigated (since you haven't corrected the resolution that leads to the contamination).**

You are right that the we use the same resolution in Rp2 as in Rp3. However, the bare ice albedo was not determined properly in Rp2, because it did not properly make use of the ice mask available, which is bug in the model. This bug is solved in Rp3, consequently increasing the bare ice albedo at the margins. To clarify this, we added the following:

**Page 8**

In Rp2, the ice mask is not projected properly on the bare ice albedo field, causing the outermost glaciated grid points to be contaminated with tundra albedo. The albedo of the outermost glaciated grid points is therefore as low as 0.3, inducing too much melt and runoff. The impact of this artifact is mitigated for higher resolutions, and is solved in Rp3 (Van Dalum et al., 2020), lowering the melt and runoff and increasing the SMB. In the southeast...

- P. 9, L. 194: Why subsurface melting of ice creates pore space? Could you more explain this affirmation and not just refer to van Dalum et al (2020). It deserves to be clearly explained, especially since it is repeated in the conclusion. This result seems counterintuitive since melt leads to an increase in density and thus to a reduction in available space.

Indeed, melt of snow and firn leads to a local thinning and not to a decrease of density. However, if ice internally is heated by radiation, it will not directly thin as the strong crystal structure inhibits compaction. Instead, small water pockets are formed, that is why melting glacial ice has a relatively high albedo. Subsequently, when in the ablation zone the winter snow has been melted away, the pore space has reduced to zero in Rp2 and the refreezing capacity and water retention capacity is zero. In Rp3, internal heating creates in the upper 20 cm of the surfacing ice some pore space. The resulting retention and refreezing capacity increase the refreezing in the ablation zone by typically 50 mm w.e. per year.

In order to take away this unclearness, Section 2.1.1 is added to describe how internal melt change subsurface layers. (See comment 4. of reviewer #2)

Furthermore, we changed the following.

**Page 9**

In Rp3, subsurface melting of glacial ice creates pore space (Sect. 2.1.1) and consequently increases the melt water retention capacity. In Rp2, no retention capacity remains once bare ice reaches the surface, limiting refreezing to the shallow winter snowpack.

**Page 21**

This increase in refreezing in Rp3 is likely realistic, as bare ice does have retention capacity for water and refreezing can occur overnight, while in Rp2 melting bare ice essentially remains dry as meltwater is removed instantaneously.

**- P. 10-11, subsection 3.2 and Figure 4: If you consider melt events (and not per year), why do you chose a threshold per year (250 mm yr-1 in the caption of the Figure 4)?**

You are right that it is strange to use a threshold in mm w.e. yr-1 here. Therefore, we now use 1 mm w.e. day-1 as a threshold. Using this rather low threshold (Fettweis et al., 2011; Fausto et al., 2016), should guarantee that all major melt events are included. The updated figures can be found at the last pages of this document.

**Page 10**

...and a minimum melt rate of 1 mm w.e. day-1 is required to be considered a melt event. Using this low threshold should guarantee that all major melt events are included (Fettweis et al., 2011; Fausto et al., 2016). All data...

- P. 10, L. 210: Results averaged over ice sheet should be given in Gt yr-1, same comment than in L. 181.

Done

Page 10 -1.1 mm w.e. yr-1 (-1.88 Gt yr-1, -0.33%) Page 10 -18.7 mm w.e. yr-1 (-32.0 Gt yr-1,-8.6%) Page 10 18.1 mm w.e. yr-1 (31.0 Gt yr-1, 12.7%)

- P. 13, L. 240: Note that R2 is the abbreviation for determination coefficient of a regression line (that expresses the part of a variable that can be expressed by another) while the coefficient correlation (that associates variations of two variables) is noted by R. Annotation (R2 to R) or the use of a non-adapted variable (correlation to determination) should also be corrected in the Table 1, Figure 5, 6 and B1 captions depending on what you meant.

We meant to use the determination coefficient, and have changed it everywhere accordingly (so in Table 1, Figure 5, 6 and B1). There was also an error in our calculations of R2 of table 1. Previously, we calculated the R2 compared to a linear regression line. However, here we use the total least square regression line (and also plot that in Fig. in 6 and Fig. B1), and now calculate R2 with respect to that. Consequently, the R2 improves compared to the previous values. We have also fixed other small mistakes, and we have updated the table and the figures accordingly. We also changed the following:

**Page 12**

The determination coefficient, root-mean-square error (RMSE) and bias with respect to observations are also shown.

**Page 13**

...show a determination coefficient (R2)...

- P.13, L. 241-243: « There is a tendency for Rp3 to underestimate LWd during cold and dry, cloudfree winter (small SWd) conditions, resulting in underestimated LWu for the same cold days. » Please could the authors clarify these explanations and what they are basing this reasoning on? Removed this sentence. We have also changed the following:

**Page 14/15**

The net longwave radiation (LWn), defined as LWd - LWu, is lower during winter for both Rp2 and Rp3, which is mostly due to the temperature bias induced by an overestimation of SHF. No changes have been made to the cloud and SHF parameterizations in Rp3, so similar results as Rp2 are therefore expected for cloudy conditions (Noel et al., 2018). Improving the representation of surface roughness may also improve these results.

**P. 14, Figure 7: Do you have an idea of why Rp3 improves SHF of Rp2 during Jul-Aug-Sep at S6? (while Rp3 has poorer results for this variable in the rest of the comparison) We added the following explanation**

**Page 15**

During summer, the SHF is overestimated by Rp2 and to a lesser degree by Rp3 for S6 (Fig. 7a). Refreezing, however, has increased for melting bare ice in Rp3 (discussed in more detail in Sect. 3.1 and Fig. 3), inducing more latent heat release, consequently heating the ice. As a result, the average skin temperature of the surface during the summer months increases and the temperature difference with the atmosphere reduces, leading to a smaller SHF. This effect is not present at S9 and S10, as bare ice rarely reaches the surface at these sites.

**The net absorbed shortwave radiation (SWn)...**

- P. 15, L. 259-261: « The introduction of a new radiation penetration and snow albedo scheme reduces the differences with observations for Rp3 compared to Rp2. This is especially noticeable at

the onset of the accumulation season [...] »: Is this amelioration statically significant? At what point is improvement considered? Do you have hypotheses to explain these discrepancy (especially in summer for SWn)? More generally, comparison between Rp2 and Rp3 (mainly Table 1 and Figure 7) suggest a relatively limited amelioration (and even deterioration of the results) poorly discussed in the text.

You are right that this explanation is poorly written. You are also right that on average, the difference between Rp3 and Rp2 for SWn is small. Therefore, we have rewritten it to the following: Page 15

The net absorbed shortwave radiation (SWn), defined as SWd - SWu, is similarly well represented by by Rp2 and Rp3 (Fig. 7, Table 1). During summer, the differences with observations increase for S6 and S9, as both model versions overestimate SWn due to a lower albedo compared to AWS observations. Note, however, that S6 is located in rough and inhomogeneous terrain, and that the local albedo and SEB may not be representative for the entire RACMO2 grid point.

Still, some differences between Rp2 and Rp3 are worth mentioning. At the onset of the accumulation season at S6, thin snow layers form and are modeled on top of bare ice. The lack of radiation penetration and internal heating in Rp2 leads to a too rapid brightening of the surface in Rp2, and subsequently underestimated SWn and hence melt-albedo feedbacks in early fall. Eventually, the new snow layer becomes thick enough to cover the bare ice even if radiation penetration is taken into account, and the differences between both model versions disappear. This effect is also to a lesser degree visible at S9. For S9, which is located close to the percolation zone, the albedo is overestimated for both Rp2 and Rp3. This is likely caused by an incorrect representation of the firn layer. The albedo differences are small for S10 (Van Dalum et al., 2020). As S10 is located in the lower accumulation zone, it is typically characterized by a homogeneous thick firn layer and little melt. Consequently, SWn is relatively similar in both model versions and is often in agreement with observations (Van Dalum et al., 2020)

- P. 16, L. 296-297: « For bands 7 to 12, almost all incoming radiation is absorbed at S6 due to the large grain radius and density. For these bands a larger fraction of energy contributes to the SEB at S6 compared to Summit. » : Figure 9 for Bands 7 to 12 at S6 suggests the opposite, the authors would probably write « a lower fraction » instead of « a larger fraction »? Yes, we meant smaller fraction, and have changed it accordingly

**- P.17, L. 308 and 309: Please specify what do you mean by « typical winter » (cloud, precipitations, ...?) and « extraordinary warm summer day ».**

We have changed the following to better explain the conditions that are shown in the figures: Page 17

First, we analyze the impact of internal energy absorption on subsurface snow temperatures. Figure 10 shows the subsurface temperature profile for Rp2, Rp3 and Rp3 WIE for S6 and Summit for a winter (Fig. 10a) and summer day (Fig. 10b). The figure also shows an example of a melt event in the accumulation zone of southeast Greenland and compares it with a melt event at S6 (Fig. 10c). All locations are shown in Fig. 2b. ...

**Page 18**

During a typical winter day (Fig. 10a, with a T2m of -11 and -18 °C respectively in Rp3), the temperature...

**Page 18**

During a typical summer day (Fig. 10b, with a T2m of 5 and -8 °C, respectively in Rp3), internal energy...

**- P. 21, L. 377: Please justify negligible.**

We have completely rewritten this section (See the last comment of Vincent Verjans)

- p. 22: Several times in the results sections, Rp3 reveals a weakness in the representation of turbulent fluxes (LHF and SHF). This deserves a point of improvement in the model in your conclusion.

...and time and would be beneficial. The turbulent fluxes could be further improved as well. We have shown that there are still some biases in the SHF and LHF, especially for areas with rough terrain like S6. A better representation of the surface roughness and the atmospheric fluxes in this complex terrain is therefore desirable (Van Tiggelen et al., 2021).

**Tables and Figures**

- Figure 2: « Subsurface temperature profiles are available for Summit » Please specify that is the green dot.

Done, but we have moved the locations to Fig. 11c, as this will now become the first figure with maps.

- Table 1: Please align the numbers to the right for better visibility ; Add unit for bias in the legend. Done

- Figure 5: Please add bias and its unit in the legend. Done

- Figure 6: Please add unit for bias in the legend. Done

- Figure 7: For more visibility, could you enlarge or thicken the coloured lines in the legend (especially for LWn, SHF and LHF)? Done

Figure 7: At certain places in the figure, the values of the three time series are similar so that it's complicated to distinguish them. This is not a problem when they are all three grouped together, but when two are grouped together and not the third one, it is difficult to distinguish which one is hidden behind the other. This can lead to a misunderstanding of the figure.
We have made the solid line a bit more transparent and the dashed line has now more space between the dashes. This way you can now distinguish all the data sets.

- Figure 10c: Please justify why this specific grid point is chosen (SE GRL, star on Figure 2). This grid point is chosen as an example of a melting conditions in the accumulation zone, which shows a different subsurface temperature profile than S6. We have added an explanation on page 17

**Page 17:**

The figure also shows an example of a melt event in the accumulation zone of southeast Greenland and compares it with a melt event at S6 (Fig. 10c).

**Editor response**

Title: I agree with the comment by Referee #2 on the current title. When I first read the title, I imagined you would discuss a radiative transfer scheme in atmosphere; however, it turned out that the main purpose of this study is to discuss impacts of a radiative transfer scheme for snow and ice. Can you specify in the title that you focus on a radiative transfer within snow and ice in this present study?

You are right that it the title might be confusing. Therefore, we have changed it to the following: Impact of updated radiative transfer scheme in snow and ice in RACMO2.3p3 on the surface mass and energy budget of the Greenland ice sheet

Sect. 6.3: I agree with Vincent Verjans that the discussion in Sect. 6.3 is a bit difficult to follow due mainly to typos. Please check and reformulate the subsection carefully. Throughout the manuscript, you explicitly indicate "snow melt". But do you really want to distinguish snow melt and ice melt in this study? I think indicating just "melt" throughout the manuscript may be enough.

As requested, we have changed 'snow melt' to 'melt' throughout the manuscript, as well as in Fig. 3 and 12. We have also rewritten Sect. 6.3, which can be found in the last comment of Vincent Verjans.

L. 23: The intention of "darkening of the snow pack" is unclear. Please clarify.

We have changed the following:

**Page 1**

..., enhancing snow metamorphism and lowering the albedo, i.e., the fraction of incoming shortwave radiation that is reflected.

L. 36: "ground" -> "subsurface conductive" Done

L. 79: Please define Bands 13 and 14 here.

Added the following: Page 3 Bands 13 and 14 (3077-3845 and 3846-12500 nm, respectively)

L. 116: "(4 minutes)" should be indicated earlier in this section, e.g., L. 100? Done

L. 107: The direction of the flux F(z) should be introduced. We added the following: Page 4

F(z) is defined positive if downwards and is then:

L. 118: More detailed explanation of "maximum skin layer equilibration depth (SLED)" is needed.

**We added the following:**

**Page 4**

...skin layer equilibration depth (SLED). In other words, the SLED is defined as the maximum depth that some energy can still equilibrate with the surface within a model time step. Using scale analysis, we estimate the SLED for both snow and ice to be 5 mm for the 4 minute time step used in this study...

L. 143: For "grain radius", you should specify whether it is geometrical or optical. This is also the case for other parts that mention snow grain size. For example, this point is particularly important when readers see Figure 8.

We changed 'grain radius' everywhere to 'optical grain radius'.

L. 147: For Rp3 WIE, you should mention whether the internal energy absorption is discarded or added to the surface; I realized that this point is explained at L. 368. But L. 147 may be the better place to explain that.

We added this:

Rp3 WIE also covers the same period of analysis as Rp3 and Rp2 and all energy is added to the SEB. Furthermore, we investigate the sensitivity of the numerical choices in the implementation of internal heating by two sensitivity experiments, in which zsled is set to 2.5 and 10 mm.

L. 177: "The SMB changes are typically significant with respect to the inter-annual variability if they exceed 20 mm w.e. yr-1.": From which figure can we see this? We have added the significance in some figures, see last comment of review #1. See also the figures added in this document.

L. 254; "net absorbed longwave radiation" -> "net longwave radiation" Done

Figure 7: It is difficult to distinguish Rp2 and Rp3 in this figure. Can you update the figure? A similar comment is provided by Referee #1. Done

L. 258: "net absorbed shortwave radiation" -> "net shortwave radiation" Done

L. 260: Please specify "the onset of the accumulation season". Page 15

At the onset of the accumulation season at the end of August or beginning of September, a thin snow layers forms on top of bare ice at S6. The lack of radiation...

Figure 8 caption: "(a), (b) Grain radius and (c), (d) absorption of solar radiation" -> "(a), (b) Grain radius and (c), (d) absorption of solar radiation simulated by Rp3" Done

L. 288: "visible light" -> "UV and visible light" Done

L. 294: "drops" -> "reductions" Done

L. 308: "temperature profile" -> "temperature profiles" Done

Figure 10 caption: "temperature profile" -> "temperature profiles" \* There are two places. Done L. 329: "temperature profile" -> "temperature profiles" Done

L. 349: Please consider adding a reference for this argument. We have added a reference to the work of Loewe (1970), with the title: "Screen Temperatures and 10m Temperatures"

L. 367: "the SMB, snow melt and refreezing difference" -> "differences in the SMB, snow melt and refreezing"; Please also check whether "snow melt" is an appropriate word or not as mentioned above.

**Done**

Page 21 Finally, we discuss the impact of radiation penetration on the SMB and analyze its components by comparing Rp3 WIE with Rp3.

L. 369: "melt" -> "meltwater" Done

L. 372 - 373: "More importantly, however, is that internal radiative heating reduces refreezing in the ablation zone (Fig. 12c), especially in the southwest, but also in the northwest and northeast (area C, D and B of Fig. 2c, respectively), so that runoff increases, reducing the SMB.": This sentence seems strange to me. Please check the sentence as well as Fig. 12 carefully. It seems to me that the SMB in the ablation area is simulated to be higher for the Rp3 than Rp3 WIE.

You are right that we did not formulate this well.

More importantly, however, is that internal radiative heating in Rp3 increases refreezing in the ablation zone (Fig. 12c), especially in the southwest, but also in the northwest and northeast (area C, D and B of Fig. 2c, respectively), reducing modeled runoff. Hence, the SMB of Rp3 WIE is lower than Rp3 for these areas.

L. 372: "area C, D and B" -> "areas A and B" We actually do mean areas C, B and D, as area A shows a different area

L. 373: "melt" -> "meltwater" Done

**L. 378 "albedo decrease" caused mainly by what?**

Added the following explanation

This region is also characterized by an albedo decrease due to slightly stronger snow metamorphism and slower firn-ice transition in Rp3 (Van Dalum et al., 2020), resulting in more...

L. 403: "resolution" -> "horizontal resolution" Done

Appendix A: The first diffusion equation contains a typo. What typo do you mean? Unfortunately, we do not see any typos.

**Interactive comment by Vincent Verjans**

For readers who are not familiar with surface energy balance and radiative energy transfer, it can be complicated to grasp all components at play. Part of the radiative energy is directly re-emitted, part of it contributes to the surface energy absorption, part of it contributes to the internal energy absorption and this partitioning depends on both wavelength and material properties. In my opinion, Figure 9 helps to how these aspects intertwine. But it comes rather late in the manuscript and it may be difficult for the reader to have a clear understanding of the radiative transfer when evaluating the results. I think it would be helpful to have a graph similar to Figure 9 alongside Figure 1, maybe for theoretical snow/ice layers and radiation conditions. Alternatively, a comprehensive table could also help. Three columns could be used ("Reflected", "Surface absorption", "Internal absorption"), with two rows ("Fresh snow", "Ice"). Each row can subsequently be split into three subrows ("UV", "Visible", "Near-IR"). The columns for each subrow can then be filled with theoretical % values or qualitative estimations ("Very low", "Low", "Medium", "High", etc.).

We have rearranged the sections of the paper such that the subsurface energy absorption section is now the first section to be discussed after the methods section (See review #1). This way Fig. 9 is discussed much earlier, which should solve most of the issues you addressed here. Fig. 9 then becomes Fig. 3.

The impact of internal energy absorption on melt in the interior should be better explained. I find the explanations provided a little contradictory: Section 3.1: " In the interior, significant differences are observed only in south Greenland, with a considerable increase of snow melt (Fig. 3a). As radiation penetration now allows internal heating, it consequently raises subsurface temperatures and increases melt." Section 6.3: " In a broad elevation band around the center, less snow melt and refreezing is observed, illustrating the importance of subsurface energy absorption, which prevents the surface from reaching the melting point."

Concerning the statement in Section 3.1, Figure 12b shows that melt differences in south Greenland are essentially albedo-driven and that the contribution of internal heating is minor (as explained at the end of Section 6.3). I understand that there might be a delicate balance between the skin temperature not reaching the melting point and extra subsurface melt. The distinction between dry and melt conditions is important for many reasons, and I hope that the authors can better quantify the isolated impact of internal energy absorption on melting in interior Greenland (which would also be crucial for large areas in Antarctica).

You are right that we could explain the processes better, therefore we change the following in Sect. 3.1 and Sect. 6.3 (Sect. 6.3 is rewritten and can be found at the last comment):

**Page 8**

...increase of melt (Fig. 3a). This is mostly caused by albedo changes, but also partly by radiation penetration inducing internal heating, which raises subsurface temperatures and increases melt, and is discussed in Sect. 6.3. Because all melt refreezes...

Figure 12b shows that internal heating increases melt in the interior. This contradicts the statement of Section 6.3. However, in some localised interior areas, internal heating decreases melt. Why do such differences occur in similar climatic conditions?

In the central area of the interior, melt events are very rare. As we show in Sect. 6.2, these rare melt events are almost always associated with cloud cover and differences are mostly due to albedo changes. However, as these melt events are so rare, these small areas with more melt can easily be caused by a small change in cloud cover between the model versions, inducing a small albedo difference and therefore also a small melt differences. By percentage, however, these changes could be quite large.

We have added a better explanation in Sect. 6.3 (Sect. 6.3 is rewritten and can be found at the last comment)

The authors have demonstrated how subsurface melt can liberate pore space in ice layers and subsequently increase refreezing capacities. This is an important point with respect to meltwater infiltration and SMB modelling. What can the authors say about the physics of that process? Is it physically plausible that subsurface melt makes ice layers more porous? Or is this simply a model artefact? I understand that the density of an ice layer affected by subsurface melt decreases in Rp3. However, could it be that it is the ice layer thickness that decreases in reality (its uppermost part melts away)? In this case, no additional porosity would be generated and runoff may be enhanced. This is a genuine question.

As stated above in reply to reviewer #1, implementing the pore space creating for ice with densities above 830 kg/m3 is a deliberate choice and not a model artefact. It should be noted that this effect occurs only in the upper 20 cm of the ice, below that point melt is negligible or ice is even below the melting point. Melting glacial or superimposed ice is slightly brittle and much brighter than pure and non-melting ice, which is an indication there are voids in the ice. Moreover, melting firn keeps more or less an equal density as the grains rearrange when they shrink or disappear due to melting. For pure ice that is impossible as the required free space to move grains around is missing. So, it is physically plausible that melting pure ice initially only gets slightly porous before it thins. It is debatable at which density grain rearrangement becomes effective and the porosity no longer increases, as far as we know, no observations are available of the density of melting glacial ice. In order to take away this unclearness, Section 2.1.1 is added to describe how internal melt change subsurface layers. (See comment 4. of reviewer #2)

Finally, please note, as is now stated in section 2.1.1, the uppermost model layer only thins irrespective of its density. We also changed this:

**Page 21**

This increase in refreezing in Rp3 is likely realistic, as bare ice does have retention capacity for water and refreezing can occur overnight, while in Rp2 melting bare ice essentially remains dry as meltwater is removed instantaneously.

Finally, here are a few technical comments. I think there might be a sign error in Figure 1. Since F(z) is decreasing with depth, should dF(z)/dz not be negative at all depth? Values on the y-axis are shown as positive however.

You are right and we have changed it accordingly.

Can the authors consider changing Figure 12 from (Rp3 WIE - RP3) to (RP3 - RP3 WIE)? This would make a comparison between Figures 3 and 12 more straightforward, so that the reader has a clearer picture of changes related to the modifications between (i) Rp2 and Rp3 and (ii) Rp3 WIE and Rp3. You are right and we have the figure from (Rp3 WIE - RP3) to (RP3 - RP3 WIE). We have updated the description of Fig. 12 and changed the text in Sect. 6.3 (See last comment for all changes in this section). We have also added the updated figure in this document

I think that "temporal variability" should be replaced by "spatial variability" on line 234. Done

I believe that there is a typo on line 297. If I understand correctly, it is a lower fraction that contributes to the SEB at S6 compared to Summit and not a "larger fraction". You are right and we have changed it accordingly.

It would be interesting to plot more than the 20 upper layers of Rp3 in Figure 10b to allow a better comparison of the model with the observations.

Unfortunately, RACMO does produce output that we can use beyond the upper 20 snow layers. Therefore, we are not able to analyze the deeper layers. We also think that the 20 upper snow layers are sufficient to discuss everything we wanted to say.

**There are some typos/repetitions in Section 6.3.**

**We have rewritten Sect. 6.3:**

Finally, we discuss the impact of radiation penetration on the SMB and analyze its components by comparing Rp3 with Rp3 WIE. Figure 12 shows the differences in the annual SMB, melt and refreezing between the two model simulations. Note that in Rp3 WIE radiation penetration is still used to determine the albedo, but all absorbed shortwave radiation is added to the SEB.

The SMB is higher for Rp3 in most of the ablation zone (Fig. 12a), in particular in the southwest (area C of Fig. 12c), where we observe a relatively small melt decrease (Fig. 12b). More importantly, however, is that internal radiative heating in Rp3 increases refreezing in the ablation zone (Fig. 12c), especially in the southwest, but also in the northwest and northeast (area C, D and B of Fig. 2c, respectively), reducing modeled runoff. Hence, the SMB of Rp3 WIE is lower than Rp3 for these areas. This increase in refreezing in Rp3 is likely realistic, as bare ice does have retention capacity for water and refreezing can occur overnight, while in Rp2 melting bare ice essentially remains dry as meltwater is removed instantaneously.

In the central part of the interior, the melt and refreezing differences between Rp3 and Rp3 WIE are smaller than for the surrounding areas (Fig. 12b, c). This supports the findings of Sect. 6.2, where we show that the albedo dominates the signal and not subsurface energy absorption. The differences between Rp3 and Rp3 WIE are consequently only small, as both model simulations use the same albedo scheme. For the surrounding areas, however, Fig. 12b and c shows that subsurface energy absorption plays a more important role, as the difference between the model simulations for melt and refreezing increases.

In south Greenland, the melt and refreezing difference between Rp3 and Rp3 WIE is smaller than between Rp3 and Rp2 (Fig. 3d). This means that processes other than subsurface energy absorption are important here. This region is characterized by an albedo decrease due to slightly stronger snow metamorphism and slower firn-ice transition in Rp3 (Van Dalum et al., 2020), resulting in more absorption of shortwave radiation, inducing melt and subsequent refreezing. Hence, it contributes to the differences between Rp3 and Rp2.

**Review # 2**

Specific comments: 1. Title: I suggest including "snow and ice" before "radiative transfer scheme" to avoid potential confusion.

Done, we changed the title to:

Impact of updated radiative transfer scheme in snow and ice in RACMO2.3p3 on the surface mass and energy budget of the Greenland ice sheet

2. Abstract (Line 6): "The surface mass balance is in good agreement with observations". Please be more quantitative, e.g., bias within a few percent?

As requested, we added the bias in the abstract, and also added the bias in the text of Sect. 3.3. Page 1

The surface mass balance is in good agreement with observations, with a mean bias of -31 mm w.e. yr-1 (-2.67%), and only changes considerably...

**Page 12**

...to observations are also shown. A slightly higher SMB is modeled for Rp3 compared to Rp2. The bias with observations for Rp3 is -0.031 m w.e. yr-1 (-2.67%) and for Rp2 -0.091 m w.e. yr-1 (-7.54%). The differences with...

3. Lines 68-71: Does the update of multilayer firn module only includes an increase in the number of layers? How about related physical processes? Any updates on the physics? More clarifications would be good.

You are right that this explanation is rather short, and have expanded it and dedicated an entire subsection to it (see next comment for the changes)

4. Lines 73-82: Although the authors mentioned that their recent paper (Van Dalum et al. 2020) provided a detailed description of the new snow albedo module, it would be good and informative to include some key elements of this new snow/ice albedo module. For example, as partially mentioned in the introduction, snow grain properties (size, shape), snow impurities, and snow layer structures are critical to snow albedo calculation. How does the new albedo module deal with these elements? Also, more descriptions of the bare ice albedo parameterization are needed. How does the module handle ice underlying snow layers, e.g. assuming a fixed ice albedo or explicitly resolve radiative transfer within ice layers below snow? Besides, what snow and ice processes have been included in this coupled RACMO2 model, e.g., sublimation, redistribution, snow grain growth, refreezing, etc?

The snow grain properties, snow impurities and etc. are all explicitly resolved using the geometricoptics theory included in TARTES, as we have mentioned in the text. To further explain what was used in Rp2, how TARTES and SNOWBAL work in Rp3 and how we handle (bare) ice and superimposed ice, we have expanded and rewritten Sect. 2.1:

**Page 3, Sect. 2.1:**

**2.1 Regional climate model**

The Regional Atmospheric Climate Model (RACMO2), developed at the Royal Netherlands Meteorological Institute (KNMI), couples the surface and atmospheric processes of the European Center for Medium-Range Weather Forecasts (ECMWF) Integrated Forecast System (IFS), cycle 33r1 (ECMWF, 2009), with the atmospheric dynamics of the High Resolution Limited Area Model, version 5.0.3 (HIRLAM, Unden et al., 2002). The polar (p) version of RACMO2, which is developed at the Institute for Marine and Atmospheric Research Utrecht (IMAU), is adjusted for glaciated areas by introducing a dedicated glaciated tile that includes snow and ice processes and more complete iceatmosphere interaction (Noel et al., 2015). Two major components have been updated in the new version Rp3 compared to the previous version Rp2: the multilayer firn module and the snow and ice albedo parameterization, which makes use of a radiative transfer scheme.

**2.1.1 Multilayer firn module**

Four modifications have been applied to the multilayer firn module, which we briefly address here and are discussed in more detail in Van Dalum et al. (2020). Firstly, Rp2 uses a prognostic fresh snow layer, which is effectively a sublayer of the uppermost snow layers. This sublayer is removed, and the uppermost snow layers are now allowed to be as thin as a few millimeters. Secondly, if merging is necessary, a snow layer now merges with its most similar adjacent layer instead of the next layer. Numerical diffusion is prevented by redistributing mass if a thin layer merges with a thick layer. Layers formed by local accumulation are not allowed to merge with glacial ice. Furthermore, the effective vertical resolution has increased, with typically 50 to 60 active snow layers, up to a maximum of 100. Model output, however, is only available for the first 20 layers. Thirdly, internal melt will thin a subsurface snow layer if the density is lower than 700 kg m-3 Pore space is created for ice layers with a density of more than 830 kg m-3. A combination of both occurs for firn with intermediate densities. Melting of the upper layer always leads to thinning. Finally, the initialized ice density is changed from 910 in Rp2 to 917 kg m-3 in Rp3.

**2.1.2 Snow albedo and radiative transfer**

The plane-parallel broadband snow albedo scheme based on Gardner and Sharp (2010), is replaced by the Two-streAm Radiative TransfEr in Snow Model (TARTES, Libois et al., 2013), which is coupled to RACMO2 with the Spectral-to-NarrOWBand ALbedo (SNOWBAL) module version 1.2 (Van Dalum et al., 2019). The broadband albedo parameterization of Gardner and Sharp (2010) is based on tuning parameters and lookup tables; and parameterizes the albedo impact of SZA, wavelength of irradiance, grain radius, cloud cover and impurities. Only the upper snow layer is considered for its calculations, with the second layer as a semi-infinite background layer. As the upper snow layers in RACMO2 are often only millimeters thick, there is virtually no radiation penetration in Rp2.

The albedo scheme of Rp3 is fundamentally different, as TARTES uses the asymptotic radiative transfer theory (Kokhanovsky, 2004; He and Flanner, 2020) and the radiative transfer equation (Jimenez-Aquino and Varela, 2005) to calculate a spectral albedo and subsurface energy absorption by using the geometric-optics method. Spectral radiative transfer in TARTES depends on snow layer density, impurity concentration, grain radius and grain shape. Here, grains are assumed to be spherically shaped; and prognostic estimates are provided for the other variables by the multilayer firn model. SNOWBAL has been developed to couple the output of TARTES with the 14 contiguous shortwave spectral bands of the IFS physics scheme embedded in RACMO2, taking into account subband variations of both the albedo and the irradiance. SNOWBAL selects a predefined representative wavelength for given atmospheric conditions suitable for TARTES to use, such that a narrowband albedo and subsurface energy flux can be accurately represented by one spectral evaluation of TARTES (Van Dalum et al., 2019). The representative wavelength depends on the SZA and vertically integrated water vapor for clear-sky conditions and ice and liquid water path for cloudy conditions. Bands 13 and 14, covering 3077-3845 and 3846-12500 nm, respectively, are excluded from calculations, as the albedo for these bands can be safely assumed to be zero (Gardner and Sharp, 2010), and all energy contributes to the SEB.

**2.1.3 Ice albedo**

Additionally, a new bare ice albedo parameterization using both TARTES and SNOWBAL has been developed. In Rp2, the bare ice albedo is prescribed by the lowest 5% of the 16 day diffuse albedo product of 1 km MODIS data (MCD43A3v5, Schaaf and Wang, 2015), resampled to the RACMO2 grid and limited between 0.3 for dark ice and 0.55 for perennial snow (Noel et al., 2018b). In Rp3, we replace these predefined bare ice albedos by using TARTES and SNOWBAL for each spectral band, allowing for varying bare ice albedo and estimates of subsurface heating. TARTES, however, is not

suitable to be applied directly to ice, as it is not based on Mie-scattering theory, and some approximations have to be made.

Firstly, we determine the specific surface area (SSA) of a semi-infinite ice layer such that TARTES and SNOWBAL calculates a broadband albedo of 0.6, which represents the broadband albedo of clean blue ice (Reijmer et al., 2001, Dadic et al., (2013) for clear-sky conditions and a reference SZA of 60 degrees. For this, we find an SSA of 0.788 m2 kg-1 (4.152 mm grain size). Then, using this SSA and SZA, a broadband albedo can be determined for a range of impurities. With this impurity range, the MCD43A3v5 MODIS albedo can be converted into an impurity concentration for each model grid point. Each time a layer is identified as bare ice, TARTES and SNOWBAL use the prescribed SSA and impurity field to do its calculations. For subsurface glacial ice layers, we use this procedure as well. Superimposed ice is treated differently than glacial ice, as it is formed by refreezing of meltwater in snow layers and has a granular structure (Granskog et al., 2006). In Rp3, superimposed ice is therefore treated as a snow layer, with a minimum grain radius of 0.720 mm for a layer density of 750 kg m-3, and the grain radius increases linearly to the bare ice value of 4.152 mm for a density of 917 kg m-3. A more detailed model description, evaluation and discussion of the snow and ice albedo product can be found in Van Dalum et al. (2020).

5. Lines 89-97: How does the model deal with the Gs (subsurface conductive heat flux) at the ice surface below snow layers? Or did the authors only consider Gs at the soil/ground surface below both snow and ice layers? Does the runoff only include water coming out of snowpack above the underlying ice? Is there lateral water flow between neighboring grids in the model? A more straightforward question would be whether the model deals with the permanent ice glaciers under snow layers? If so, how?

At each glaciated grid point, a snow/firn/ice column of at least 30 m and 30 layers is considered, of which at least the lowermost one consists of glacial ice. If ablation would reduce the thickness to a value less than 30 m, the lowermost model layer is duplicated. The temperature of the lowermost layer is kept constant at its initialized value and with the exception of points with strong ablation, this lowermost temperature is effectively irrelevant. At these glaciated points, the normal soil scheme is not considered, and there is no interaction between the soil and the glacial ice.

Gs is thus the energy flux between the skin layer, the contact layer between the atmosphere and the surface, and the uppermost layer of the firn column. Between all model layers, Gs is derived and used for the subsurface temperature evolution, but this flux is not stored. Meltwater runoff is directly from the firn/ice column without interference of the normal soil layers. There is no lateral water flow between neighboring glaciated cells.

**Page 4**

... or runs off. There is no lateral flow between grid points. The subsurface conductive heat flux Gs is the energy flux between the skin layer, i.e., the contact layer between the atmosphere and the surface, and the uppermost layer of the firn column. Between all model layers, Gs is also derived and used for the subsurface temperature evolution, but this flux is not stored.

6. Section 2.3: The authors calculated internal energy absorption at non-FR time steps by using a simplified Beer's law (i.e., exponential decay) equation. Would this cause any discontinuity of the absorption profiles between non-FR and FR time steps since the TARTES is used for FR time steps? To prevent discontinuities of the absorption profiles on non-FR time steps, we introduced the attenuation length tau, which we explained in Sect. 2.3. With this term, we transformed the subsurface energy flux as determined by TARTES on a FR time-step to a quantity that could be used on a non-FR time step. So, at the FR time steps the two methods give, by definition, equivalent heating rates. On each non-FR time step, RACMO calculates an incoming shortwave radiative flux at the surface. Using this tau, one can then distribute the energy among the layers accordingly without any discontinuity even if layer thicknesses have changed.

**7. Section 2.5: What are the observational uncertainties of these in-situ measurements?**

We have added the following:

**Page 7**

...but are mostly located in the ablation zone. A first order approximation of the uncertainty of the SMB observations according to Machguth et al. (2016) is 0.2-0.4 m w.e. yr-1. AWS data of both... Page 8

For the K-transect, this is discussed in more detail by Smeets et al. (2018). Smeets et al. (2018) also report an uncertainty for shortwave radiation of approximately 1%, for longwave radiation of  $\pm 5$  W m-2 for T2m of max  $\pm 0.5$  degrees C, but usually  $\pm 0.2$ -0.3 degrees C and for V10m of at least  $\pm 0.2$  m s-1

8. Section 3.2: This part is very interesting, however, I am not quite convinced that the rather small snow melt in the central GrIS could have such a large impact on the cloud cover in the region. Any explanations on the mechanisms? Did the authors see an increase in surface heat and water vapor sources due to snow melting and albedo increase, which could enhance the cloud formation locally? Also, snow melting could also decrease snow albedo, so I am not sure if the change in spectral distribution of irradiance is large enough to cause the snow albedo increase as mentioned by the authors. More explanations and clarifications are needed.

Unfortunately, you misunderstood our explanations. We tried to say in Sect 3.2 that due to clouds, the solar radiation that reaches the surface is shifted more towards the visible light, away from infrared. For visible light, the snow albedo is higher. In Rp3, this effect is now considered, so the albedo is expected to increase with respect to Rp2. So, we do not claim that snow melt increases the cloud cover in the region. We try to explain that during the rare melt events in central Greenland, there is almost always cloud cover, which is now associated with a higher albedo (Fig. 4). Then, due to this higher albedo, we show that the snowmelt is reduced (Fig. 3). This is only true for the central region of Greenland, for the other regions other processes are more important, which are discussed in this manuscript as well. As for the impact that the change of spectral distribution of the irradiance has on the albedo, you can look at our previous work (Van Dalum et al., 2019, 2020), but also at Gardner and Sharp (2010), Dang et al., 2015, Warren (2019).

We added these references and added a few sentences to hopefully better explain what we mean. Page 11

...for which the albedo is low (Dang et al., 2015; Van Dalum et al., 2019; Warren, 2019).

**Page 11**

...less melt is modeled. To summarize, cloud cover during melt events changes the spectral distribution of shortwave radiation at the surface, shifting more towards shorter wavelengths (UV and visible light). As the albedo is higher for these shorter wavelengths, which is now properly modeled in Rp3, less energy is available for melt. Note that the...

9. The authors conducted model evaluation by comparing with several in-situ point measurements. I suggest including some regional scale evaluation of the model, particularly surface albedo, by comparing with satellite products (e.g., MODIS). Large uncertainty could exist in the comparison of 11-km model grid data with point observations.

It is a fair point to make that the albedo needs evaluation as well. This is already done, however, in the paper published last year in The Cryosphere by C.T. van Dalum, W.J. van de Berg, S. Lhermitte and M.J. van den Broeke, with the title: "Evaluation of a new snow albedo scheme for the Greenland ice sheet in the Regional Atmospheric Climate Model (RACMO2)" (DOI: 10.5194/tc-2020-259). It also includes a regional scale evaluation and a MODIS comparison.

**Revised figures**

The following figures have been updated. We have also increased the resolution of most figures. Fig 1. Fig 2